Manuscript prepared for Atmos. Chem. Phys. Discuss.
with version 2015/04/24 7.83 Copernicus papers of the LaTeX class copernicus.cls.
Date: 6 July 2017

# Regional effects of atmospheric aerosols on temperature: an evaluation of an ensemble of on-line coupled models

**Rocío Baró**[1,2]**, Laura Palacios-Peña**[1]**, Alexander Baklanov**[3]**, Alessandra Balzarini**[4]**, Dominik Brunner**[5]**, Renate Forkel**[6]**, Marcus Hirtl**[2]**, Luka Honzak**[7]**, Juan Luis Pérez**[8]**, Guido Pirovano**[3]**, Roberto San José**[8]**, Wolfram Schröder**[9]**, Johannes Werhahn**[5]**, Ralf Wolke**[9]**, Rahela Zabkar**[10]**, and Pedro Jiménez-Guerrero**[1]

[1]Department of Physics, Regional Campus of International Excellence Campus Mare Nostrum, University of Murcia, Murcia, Spain
[2]Section Chemical Weather Forecasts, Division Data/Methods/Modelling, ZAMG - Zentralanstalt für Meteorologie und Geodynamik, Austria
[3]World Meteorological Organization, Geneve, Switzerland
[4]Ricerca sul Sistema Energetico (RSE), Italy
[5]Laboratory for Air Pollution/Environmental Technology, Empa, Swiss Federal Laboratories for Materials Science and Technology, Switzerland
[6]Karlsruher Institut für Technologie (KIT), Institut für Meteorologie und Klimaforschung, Atmosphärische Umweltforschung (IMK-IFU), Germany
[7]BO-MO d.o.o, Slovenia
[8]Environmental Software and Modelling Group, Computer Science School - Technical University of Madrid, Spain
[9]Leibniz Institute for Tropospheric Research, Permoserstr, Germany
[10]Slovenian Environment Agency, Slovenia

Correspondence to: Pedro Jiménez-Guerrero (pedro.jimenezguerrero@um.es)

**Abstract**

The climate effect of atmospheric aerosols is associated to their influence on the radiative budget of the Earth due to the direct aerosol-radiation interactions (ARI) and indirect effects, resulting from aerosol-cloud-radiation interactions (ACI). On-line coupled meteorology-chemistry

models permit the description of these effects on the basis of simulated atmospheric aerosol concentrations, although there is still some uncertainty associated to the use of these models. In this sense, the objective of this work is to assess whether the inclusion of atmospheric aerosol radiative feedbacks of an ensemble of on-line coupled models improves the simulation results for maximum, mean and minimum temperature at 2 meters over Europe. The

evaluated models outputs originate from EuMetChem COST Action ES1004 simulations for Europe, differing in the inclusion (or omission) of ARI and ACI in the various models. The cases studies cover two important atmospheric aerosol episodes over Europe in the year 2010, a heat wave event and a forest fires episode (July-August 2010) and a more humid episode including a Saharan desert dust outbreak in October 2010. The simulation results

are evaluated against observational data from E-OBS gridded database. The results indicate that, although there is only a slight improvement in the bias of the simulation results when including the radiative feedbacks, the spatio-temporal variability and correlation coefficients are improved for the cases under study when atmospheric aerosol radiative effects are included.

**1   Introduction**

Atmospheric aerosol particles are known to have an impact on Earth's radiative Budget due to their interaction with radiation and clouds properties, which is dependent on their optical, microphysical and chemical properties, and are considered to be the most uncertain forcing agent. They influence climate by modifying the global energy balance through both

absorption and scattering of radiation (direct effect) and by acting as cloud condensation nuclei, thus affecting clouds droplet size distribution, lifetime (Twomey, 1977; Lohmann and

Feichter, 2005; Chung, 2012) and reflectance (indirect effects) (Ghan and Schwartz, 2007; Yang et al., 2011). Depending on the atmospheric aerosol concentration, aerosol cloud interactions may result in an increase or decrease in liquid water content, cloud cover, and lifetime of low level clouds and a suppression or enhancement of precipitation (Bangert et al., 2011). Besides, aerosol absorption may decrease low-cloud cover by heating the air and reducing relative humidity. This leads to a positive radiative forcing, termed the semi-direct effect, which amplifies the warming influence of absorbing aerosols (Hansen et al., 1997). The Fifth Report of the Intergovernmental Panel on Climate Change (IPCC AR5) (Boucher et al., 2013; Myhre et al., 2013) distinguishes between aerosol-radiation interactions (ARI), which encompass the aerosol direct and semidirect effect, and the aerosol-cloud interactions (ACI), which encompass the indirect effects.

In order to account for these atmospheric aerosol effects, the use of fully-coupled models is needed for meteorological, chemical and physical processes. On-line coupled models include the interaction of atmospheric pollutants (gaseous-phase compounds and aerosols) with meteorological variables (Baklanov et al., 2014). In this context, in its phase 2, the air quality model evaluation international initiative (AQMEII) (Alapaty et al., 2012; Galmarini et al., 2015) focused on the assessment of how well the current generation of coupled regional scale air quality models can simulate the spatio-temporal variability in the optical and radiative characteristics of atmospheric aerosols and associated feedbacks among aerosols, radiation, clouds and precipitation. On this basis, a coordinated exercise of working groups 2 and 4 of the COST Action ES1004 European framework for online integrated air quality and meteorology modelling (EuMetChem, http://eumetchem.info) emerged in order to take into account the radiative feedbacks of atmospheric aerosol effects on meteorology. In this initiative, two important episodes with high loads of atmospheric aerosols were analyzed which were identified during the previous AQMEII phase 2 modelling intercomparison exercise (Galmarini et al., 2015). They were selected on behalf of their strong potential for aerosol-radiation and aerosol-cloud-radiation interactions (Makar et al., 2015a, b; Forkel et al., 2015).

As a result of the AQMEII phase 2 initiative and EuMetChem COST Action, several studies covering the analysis of the ARI+ACI feedbacks to meteorology have been done (e.g. Baró et al. (2015); Forkel et al. (2015, 2016); Kong et al. (2015); San José et al. (2015)). Focusing on the effects of including ARI+ACI on temperature, Forkel et al. (2015) focused
on the 2010 Russian wildfire episode, where the presence of the atmospheric aerosols decreased the 2-m mean temperature during summer 2010 by 0.25 K over the target area. For the same episode, Péré et al. (2014) showed daily mean surface temperature reductions between 0.2 to 2.6 K. In Forkel et al. (2012) they studied a two-month episode (June to July 2006) for allowing medium range effects of the direct and indirect aerosol effect on mete-
orological variables and air quality. They found a slightly lower temperature over western Europe when including atmospheric aerosol feedbacks. This reduction followed the same pattern as the planetary boundary layer height. Moreover, Meier et al. (2012) found during July 2006 a general decrease of 0.14 K on 2-m temperature when simulating absorbing aerosol in upper layers compared to an aerosol-free troposphere over land surface.
However, all these studies are based on individual model evaluations and do not take into account an ensemble of regional models, in order to build confidence on model simulations and to characterize the uncertainty associated to the use of different modelling systems. Therefore, the objective of this work is to assess whether the outputs of an ensemble of regional on-line coupled model simulations including aerosol radiative feedbacks, during
two important atmospheric aerosol episodes of the year 2010, improves the prognostic for maximum, mean and minimum temperature at 2 meters over Europe.

## 2 Methodology

The analyzed model outputs are the results of a coordinated modelling exercise which was performed within the COST Action ES1004 (EuMetChem). In order to analyze the ARI or
25 ARI+ACI effect on temperature, it was suggested to run three case studies for two episodes with different on-line coupled models with identical meteorological boundary conditions and anthropogenic emissions. The two considered episodes are: the Russian heatwave and

wildfires episode in the summer of 2010 (25 July-15 August 2010) and an autumn Saharan dust episode, including the dust transport to Europe (2-15 October 2010).

The weather conditions during the Russian forest fires were mainly dry and particularly hot, with light winds (Péré et al., 2014; Kong et al., 2015). During this situation, the sea-level pressure (SLP) showed a high-pressure system over the northeast part of the Russian area, finding a strong positive SLP anomaly for this period. This resulted in a strong positive surface temperature anomaly accompanied by weak winds from the southeast (Baró et al., 2017). On the other hand, the dust period situation is characterized by a very deep trough with a vortex reaching 20$^o$N latitude. This situation is maintained for several days, causing a continuous transport in middle levels. It is also worth mentioning the blocking situation over all central Europe. The dust event was dominated by strong south-easterly wind. This may explain windblown dust emissions increasing with wind speed and being transported to some parts of the European area (Kong et al., 2015).

For the chosen episodes, simulations with each model were performed with and without considering the atmospheric aerosol effects. Three different configurations were requested: the first one which does not consider any aerosol effects feedbacks to meteorology (NRF; C11 fire and C21 dust episode); second, where only aerosol-radiation interactions are considered (ARI; C12 fire and C22 dust episode) and third, where aerosol-radiation-cloud interactions are considered (ARI+ACI; C13 fire and C23 dust episode)(this case could not be submitted by all of the participants). Although the NRF case does not consider the aerosol effects and feedbacks, this configuration considers an assumption of 250 cm$^{-3}$ used by WRF-Chem in the absence of ACI for estimating cloud droplet number. This number is used in the corresponding microphysics parameterization (Morrison or Lin). On the other hand, ARI uses this constant value for accounting the interaction between aerosols and clouds, but allows the modification of the radiation budget by using the on-line estimated aerosols. Last, the ARI+ACI cases are based on simulated aerosol concentrations which interact both with radiation and aerosols. The common setup for the participating models and a unified output strategy allow analyzing the model output with respect to similarities and differences in the model response to the aerosol direct effect and aerosol-cloud interactions.

## 2.1 Participating models

An overview of the different models and their configurations is shown in Table 1, where in first row the model acronym is shown. The participating models shown here are COSMO-MUSCAT (Wolke et al., 2012) and WRF-Chem (Grell et al., 2005; Fast et al., 2006; Gustafson Jr

et al., 2007; Chapman et al., 2009; Grell and Baklanov, 2011) with different chemistry and physics options. The table also includes the episodes run for each model. The horizontal grid spacing is around 25 km for most of the contributions. Only for the fire episode, the COSMO-MUSCAT simulations were made with a grid with of 0.125 deg (approximately 14 km) there is an additional WRF-Chem run with 9 km grid spacing. The COSMO models use

Kessler-type bulk microphysics (Doms et al., 2011) and WRF-Chem uses Morrison microphysics (Morrison et al., 2009), except for one contribution, that utilizes Lin (Lin et al., 1983). COSMO models use prognostic TKE (Doms et al., 2011) planetary boundary layer (PBL). The YSU PBL scheme (Hong et al., 2006) was chosen for the WRF-Chem simulations. In general, the Modal Aerosol Dynamics Model for Europe (MADE) is applied (Ackermann

et al., 1998) except for one WRF-Chem simulation, which uses the Model for Simulating Aerosol Interactions and Chemistry (MOSAIC)(4 aerosol size bins) approach (Zaveri et al., 2008). For further information and details about the models, we refer to the work of Forkel et al. (2015); Im et al. (2015a, b); Baró et al. (2015). To enable the cross-comparison between models, the participating groups interpolated their model output to a common grid

with 0.1 degree resolution.

Moreover, the ensemble of the available simulations has also been included in this comparison, as recommended by several studies (Vautard et al., 2012; Jiménez-Guerrero et al., 2013; Landgren et al., 2014; Solazzo and Galmarini, 2015; Kioutsioukis et al., 2016), in order to check whether the design of an ensemble of simulations outperforms (or not) the skill

of individual models.

## 2.2 Emissions and boundary conditions

For the EU domain, the anthropogenic emissions for the year 2009 (http://www.gmes-atmosphere.eu/) were applied by all modelling groups and are based on the TNO-MACC-II (Netherlands Organization for Applied Scientific Research, Monitoring Atmospheric Composition and Climate–Interim Implementation) framework (Kuenen et al., 2014; Pouliot et al., 2015). As described in Im et al. (2015a), annual emissions of methane ($CH_4$), carbon monoxide (CO), ammonia ($NH_3$), total non-methane volatile organic compounds (NMVOC), nitrogen oxides ($NO_x$), particulate matter (PM10 & PM2.5) and sulfur dioxide ($SO_2$) from ten activity sectors are provided on a latitude/longitude grid of $1/8 \times 1/16$ resolution. Consistent temporal profiles (diurnal, day-of-week, seasonal) and vertical distributions were also made available to AQMEII and EuMetChem participating groups for time disaggregation. The temporal profiles for the EU anthropogenic emissions were provided from Schaap et al. (2005). For further details, the reader is referred to Im et al. (2015a, b).

Hourly biomass burning emissions were provided by the Finnish Meteorological Institute (FMI) fire assimilation system (http://is4fires.fmi.fi/) (Sofiev et al., 2009). More details on the fire emissions and their uncertainties are discussed in Soares et al. (2015). The fire assimilation system provides only data for total PM emissions; the estimation of emissions for other species are described in Im et al. (2015b).

The chemical initial conditions (IC) were provided by the European Centre for Medium–Range Weather Forecasts (ECMWF) IFS–MOZART model, which are available in 3–hour time intervals and provided in daily files with 8 times per file. They were run under the MACC-II project (Monitoring Atmospheric Composition and Climate – Interim Implementation) which uses an updated data set of anthropogenic emissions and compiles a satellite observations assimilations of $O_3$, CO and $NO_2$ in the IFS-MOZART system.

## 2.3 Observational database

The comparison of regional models with gridded datasets has to be carefully taken into account given the differences between available databases. For instance, Gómez-Navarro

et al. (2012) showed that even in areas covered by dense monitoring networks, uncertainties in the observations are comparable to the uncertainties within state-of-the-art regional climate models, at least when they are driven by nominally perfect boundary conditions like reanalysis.

This work uses the E-OBS (Haylock et al., 2008) version 11.0 gridded observational database for maximum, mean and minimum temperature. E-OBS is a high-resolution European land-only daily gridded data set covering the period 1950-2014. The E-OBS 0.25 degrees regular latitude-longitude grid has been used as the reference for validation. Thus, data from all model runs have been bilinearly interpolated onto the E-OBS grid. Since the
resolution of the models is similar to that of E-OBS, the interpolation procedure is not expected to alter significantly our results.

    The election of this gridded dataset is based on the abundant scientific literature using E-OBS for the evaluation of regional climate models (e.g. Costa et al. (2012); Jiménez-Guerrero et al. (2013); Turco et al. (2013); Ceglar et al. (2014), among many others). How-
15 ever, several authors highlight the E-OBS limitations. In this sense ,Kysely and Plavcova (2010) compare E-OBS and a data set gridded onto the same grid from a high-density network of stations in the Czech Republic (GriSt), finding that large differences existed between the two gridded data sets, particularly for minimum temperatures and diurnal temperature range. The errors tended to be larger in tails of the distributions. Therefore, when evaluating
regional models against one gridded dataset, results have been to be carefully taken into account.

## 2.4  Validation methodology

All the statistical measures are calculated at individual grid points. Only land grid points are considered in the analysis, since these are the only points where E-OBS contains informa-
25 tion. Areas in grey indicate cells where E-OBS data are not available (southeastern part of the domain for the wildfires or southern part of the domain in the dust episode) or areas not covered by the modelling domain (southern part of the domain for the CS2 configuration).

We will use the notation $V_{ipc}^k$ for a variable from model $k$ at grid point $i$, on period p=fires,dust and case c=1 2, 3 representing no radiative feedbacks, ARI and ARI+ACI. If we use bracket notation for an average over a given index (e.g. $\langle \cdot \rangle_{pc}$, we can express the bias at a given grid point as:

$$5 \quad \mathsf{b}_i^k = \left\langle V_{ipc}^k - O_{ip} \right\rangle_{pc} \tag{1}$$

where $O_{ip}$ is the value observed. The model bias is the simplest measure of model performance.

The ensemble mean, $\left\langle V_{ipc}^k \right\rangle_k$, is usually considered as an additional simulation which compensates the errors of the different ensemble members. Even though this is a very simplistic view of the ensemble (which should be considered from a probabilistic point of view), it can be useful to reinforce the common signal of the different models in our analysis of the mean climate. Notice, however, that the ensemble mean is not a physical realization of any of the models, but just a statistical average (Knutti et al., 2010; Jiménez-Guerrero et al., 2013).

Then, the variability was assessed on the hourly series ($V_{ipc}^k$). The ability to represent the variability can be decomposed into:

- the ability to represent its size, which can be represented by the standard deviation of the series:

$$\mathsf{sd}[V]_i^k = \sqrt{\left\langle \left( V_{ipc}^k \right)^2 \right\rangle_{pc}} \tag{2}$$

and can be compared to that of the observations $\mathsf{sd}[O]_{ip}$, and

**–** the ability to represent the hourly variations, which can be represented by the linear determination coefficient ($\rho^2$) with the observations.

$$\rho_i^{2,k} = \frac{\left\langle V_{ipc}^k O_{ip} \right\rangle_{pm}^2}{\left\langle \left(V_{ipc}^k\right)^2 \right\rangle_{pm} \left\langle (O_{ip})^2 \right\rangle_{pc}} \tag{3}$$

The latter ability can only be expected on simulations nested into "perfect" boundary condi-
5 tions such as those considered in this study.

Finally, pattern agreement between simulated and observed data was quantified in a Taylor diagram by means of the spatial correlation (r) and the ratio between simulated and observed standard deviations, $V_i^k \equiv \left\langle V_{ipc}^k \right\rangle_{pc}$

$$r^k = \frac{\left\langle \left(V_i^k - \left\langle V_i^k \right\rangle_i\right) \left(O_i - \left\langle O_i \right\rangle_i\right) \right\rangle_i}{\sqrt{\left\langle \left(V_i^k - \left\langle V_i^k \right\rangle_i\right)^2 \right\rangle_i \left\langle \left(O_i - \left\langle O_i \right\rangle_i\right)^2 \right\rangle_i}} \tag{4}$$

$$s^k = \sqrt{\frac{\left\langle \left(V_i^k - \left\langle V_i^k \right\rangle_i\right)^2 \right\rangle_i}{\left\langle \left(O_i - \left\langle O_i \right\rangle_i\right)^2 \right\rangle_i}} \tag{5}$$

This information can be summarized in a Taylor (2001) diagram, which is a polar plot, with radial coordinate $s^k$ and angular coordinate related to $r^k$.

## 3 Results

15 ### 3.1 Aerosol representation

In order to address the influence of aerosols effects on the surface temperature it is cru-
cial to have an understanding of the aerosol loading, both observed and modelled. For that

purpose, aerosol optical depth (AOD) from MODIS platform (Levy et al., 2013) is used, precisely Level 2 of Atmospheric Aerosol Product (MxD04_L2), collection 6 (C6) with a resolution of 10 km. Palacios-Pena et al. (submitted to ACP, this issue) provided full details of the evaluation of the same set of models presented here against diverse satellite observa-
tions for AOD. The current contribution includes a brief description of the results. Figure 1 represents the Model-MODIS comparison of AOD at 550 nm both for the fires and the dust episode.

For the Russian wildfires episode, the highest values of AOD measurements by MODIS (around 2.7) are found over Russia and surroundings areas, due to the emissions produced
by the wildfires. According to the estimation of the bias (MBE), all WRF-Chem simulations (CS1, CS2, ES1, ES3) and the ensemble underestimate AOD over the fire-affected areas (minimum MBE values for NRF: the ensemble -1.30; CS1 -1.46; CS2 -1.61; ES1 -1.46 and ES3 -1.62). Over the rest of the domain, a lower overestimation (around 0.5) is produced by the WRF-Chem simulations (maximum MBE values for NRF: CS1 0.55; CS2 0.37; ES1
0.45 and ES3 0.64) and the ensemble (maximum MBE values for NRF 0.23). For DE3, the underestimation is lower (minimum MBE values for NRF -0.72) and does not cover a so large area as the rest of the experiments; however, over the rest of the domain a higher overestimation is found in DE3 (maximum MBE values for NRF 2.61). Generally, for ARI and ARI+ACI simulations, slightly lower MBE values than NRF are found in all the experiments
(for example, in ES1 simulations: NRF -1.46; ARI -1.43; ARI+ACI -1.41). However, the MBE for the ensemble (NRF -1.3; ARI -1.23; ARI+ACI -1.40) does not show this improvement; but his analysis should be carefully taken into account because the ARI+ACI ensemble does not include DE3 simulations.

For the dust episode, AOD values measured by MODIS $> 0.5$ are observed over the
southeast of the domain due to the dust transported. This value is not very high for a dust outbreak, but this is caused by the wet deposition (rain during the episode). The highest AOD values, around 1.3, are found over a small area near the Po Valley. All experiments (no CS2 simulations are available in this case) underestimated high AOD values (over the southeast of the domain). MBE values over this area are around -0.3 for DE3 but for the

rest of the experiments (WRF- Chem simulations) these values are around -0.2. However small areas with a higher underestimation are found over this zone (minimum MBE values: the ensemble -0.73; CS1 -0.68; DE3 -0.84; ES1 -0.70; ES3 -0.67). Over the rest of the domain, small overestimations are modeled (MBE values around 0.1). Conversely, small punctual areas with a high overestimations are found (maximum MBE values for ENS 0.54; CS1 0.81; DE3 0.62; ES1 0.48; ES3 1.09).

### 3.2 Bias

The results for the daily bias of maximum, mean and minimum temperature have been obtained by calculating the bias of the daily mean series at each grid point of all the land grid points of the corresponding domain for the fires and dust episodes. They are summarized in Table 2 for the entire domain. Table 3 only considers the biases in those cells and timesteps with a high load of aerosols (masking only those areas where 1-hr AOD550>1.0 in the fires case or 1-hr AOD550>0.5).

During the fire episode (Fig. 2 left column) there is a general underestimation of the maximum temperature in the base case (average domain values from -2.1 K in ES3-C11 to -1.2 K in DE3-C11 for the entire domain; or -5.7 K in CS2-C12 to -3.0 K in DE3-C11 only in those areas with 1-hr AOD550>1.0). This is especially noticeable over several cells in Russia (up to -7 K). Conversely, a general overestimation is found in the west and northwest area of the domain (positive differences between +1.0 K in DE3-C11 to +6.5 K in ES1-C11). When introducing the ARI or ARI+ACI, model biases do not improve (mean variation of the bias of +17.2% in C12 and +11.0% in C13 for the entire domain). This positive variation was expected because the cold bias of models for reproducing maximum temperature and the overall cooling effects of aerosols. However, the improvement of introducing aerosol-cloud interactions is remarkable with respect to the case of including just aerosol-radiation effects (the bias reduces 6.2% in ARI+ACI with respect to ARI simulations). During the dust episode (Fig. 2 right column) the analysis of the results is similar as for the fires case (averaged-domain underestimations around -1.0 K in DE3-C11 to -0.56 K in ES1-C21; -4.1 K in DE3-C21 to -2.8 K in ES1-C11 only for areas and timesteps where 1-hr AOD550>0.5).

Here the inclusion of ARI (C22) leads to a mean increase of the bias of +10.2% for the entire domain, but ARI+ACI (C23) leads to a very limited improvement of the simulations with respect to the base case (C21), generally reductions of the bias around -0.4%.

A similar discussion can be made for mean temperature. During the fires episode (left column of Fig. 3) all runs (but DE3) tend to underestimate the domain-averaged mean temperature (biases ranging from -0.4 K in ES1-C11 to +1.0 in DE3-C11; for those areas when AOD550>1.0 biases range from -1.1 K in CS2-C13 to +1.0 in DE3-C11). Here, the ensemble (ENS) simulation clearly outperforms the individual simulations (bias of -0.2 K in ENS-C11 for the entire domain and -0.1 K in the high-AOD domain). Again, the model skill does not improve for mean temperature when including ARI or ARI+ACI (bias increase by 46.0% and 56.2%, respectively for the fires episode averaged over the entire domain) but in the case of DE3-C12 simulation (including ARI reduces the bias by -27.3%). During the dust episode (right column of Fig. 3), there is a general averaged overestimation of mean temperature (+0.4 in ES1-C21 to 0.8 K in DE3-C21; for those areas when AOD550>0.5 biases range from -0.5 K in CS1-C21 to -0.1 in ES1-C11). Conversely to the fires episode, the inclusion of ARI and ARI+ACI improves the entire-domain bias (reductions of this variable of -13.4% in C22 and -4.2% in C23). The reduction of the bias when including ARI+ACI is especially remarkable for the ensemble of simulations, where the bias decreases by -24.4% in ENS-C23 (averaged for the entire domain).

Last, minimum temperature during the fire episodes is shown in the left column of Fig. 4. Here results are very different to analyze for improvements or worsening of the bias, since the domain-averaged errors are in the order of -0.01 K for WRF-based models in C11 and C12, so a very slight difference would lead to a percentage increase (or reduction) of the bias compared to the base case. However, for DE3-C11 the bias is larger (up to +1.6 K for minimum temperature averaged over all the domain) and the inclusion of ARI leads only to a small improvement (-1.5%). Despite the conclusions are similar for for areas with 1-hr AOD550>1.0, WRF-Chem based models present biases around +3.0 to 3.5 K for the fires episode; and around +4.5 K for DE3-C11 and DE3-C12. The dust case (right column of Fig. 4) shows a general overestimation of minimum temperature for domain-averaged values,

with base-case biases ranging from +0.5 K in ES1-C21 to +1.8 K in DE3-C21 (biases from +2.0 in ES3-C23 to +3.5 in DE3-C21 in areas with AOD550>0.5). Here, the inclusion of ARI and ARI+ACI slightly improves the average bias for the entire domain (reductions of -10.5% in C22 and -5.0% in C23). Here again, the improvement of the ENS-C22 and ENS-C23 simulations is larger than for the rest of the models (reductions of the bias of -29.7% and -38.2% for ARI and ARI+ACI, respectively). Analogous discussions can be done for the masked domain according to the AOD550 values.

## 3.3 Temporal correlation

The temporal correlation (estimated through the coefficient of determination, $\rho^2$) between simulated and observed series is shown in Fig. 5, 6 and 7 for mean maximum and mean minimum temperature, in that order. They are also summarized in Table 4 for the entire domain. Table 5 only considers the temporal correlation in those cells and timesteps with a high load of aerosols (masking only those areas where 1-hr AOD550>1.0 in the fires case or 1-hr AOD550>0.5). Since the values and conclusions are very similar, only the results from the entire domain are discussed below.

The first column in each panel represents the value of $\rho^2$ of the base case (C11 or C21) of each individual model (or the ensemble) with respect to the E-OBS database. The center (C12 or C22) and right (C13 and C23) columns indicate the increase (red values) or decrease (blue value) of the $\rho^2$ for each simulation with respect to the case not including feedbacks. Then, that gives an idea in the improvement (or not) in the skill of the model for representing the time evolution of our series when compared to the observations.

For maximum, mean and minimum temperature during the fires episode (left side of Fig. 5, 6 and 7, respectively), domain-averaged $\rho^2$ is higher than 0.5 for all models (0.52 in CS1-C11 minimum temperature to 0.78 in DE3-C11 mean temperature). In general, coefficients of determination are highest for mean temperature (ranging from 0.60 to 0.78 depending on the individual model) with respect to minimum and maximum temperature. The variable with the lowest $\rho^2$ is minimum temperature (varying from 0.50 to 0.56 depending on the model). Moreover, the coefficient of determination for the ensemble is always higher than

that of each individual models for the three studied variables (0.75, 0.79 and 0.61, respectively for maximum, mean and minimum temperature). The highest $\rho^2$ values are found over the north and west part of the domain (above 0.8 in mean temperature) and the lowest mainly over south and southeast area of the domain (under 0.2). According to the improvement with respect to C11 case, when analyzing the inclusion of the ARI and ARI+ACI a general improvement is observed for maximum and mean temperature, with positive values reaching up to 0.18 (domain-averaged values improve for individual models around 1% for maximum, 0.3% for mean temperature). Correlation with minima experiences a slight decrease (-0.4%) when including ARI or ARI+ACI for the ensemble mean.

During dust episode (right side of Fig. 5, 6 and 7), domain-averaged $\rho^2$ is higher than for the fires episode for all models and variables in the base case (0.76 in DE3-C21 minima to 0.90 in DE3-C21 mean temperature), with the ensemble again providing the highest correlation (values ranging from 0.88 for maximum, 0.91 for mean and 0.84 for minimum temperature). As well as before, the inclusion of the ARI and ARI+ACI shows an improvement over some areas in the order of 0.17 for mean and maximum temperature, with domain-averaged improvements of 0.3% in C22-C23 for maximum temperature, and 0.2% in C22-C23 for mean temperature and 0.1% in C23 for minimum temperature, with no improvement for C22 in this latter variable).

## 3.4 Temporal variability

The results for the daily variability of maximum, mean and minimum temperature have been obtained by calculating the standard deviation of the daily mean series at each grid point of all the land grid points of the corresponding domain for the fires and dust episodes.

Considering maximum temperature, in the fires episode (left column of Fig. 8), all runs tend to slightly overestimate the standard deviation of maximum temperature for the base case (no radiative feedbacks), with biases of maximum temperature standard deviation varying between +1.28 K for DE3-C11 to +0.25 K for ES1-C11. The biases of the standard deviation are reduced by -22.6% (on average) when including the ARI, with reductions in the biases of the standard deviation ranging from -34.2% in ES1-C12 and -8.6% for DE3-

C12. For the ARI+ACI simulations the average reduction of the bias is -41.21% (-56.9% for ES1-C13 and -24.40% for CS2-C13). The rest of the models and cases show an intermediate behavior for representing the variability, with the best skills always for the cases including the ARI+ACI interactions. Analogous results can be found for maximum temperature during the dust episode (right column of Fig. 8): the inclusion of aerosol feedbacks generally improve the representation of the temporal variability of maximum temperature, with an average reduction of the bias of the standard deviation of -5.9% (-16.6%) for ARI (ARI+ACI) simulations.

For mean temperature during the fires episode, (left column of Fig. 9) all runs tend to overestimate the standard deviation for the base case (no radiative feedbacks), with biases of mean temperature standard deviation between +0.2 to +1.1 K. As for the maximum temperature, the biases of the standard deviation are reduced on -41.8% (on average) when including the ARI and -66.5% for the ARI+ACI simulations, with reductions in the biases of the standard deviation ranging from -8.5% in the DE3-C12 simulation to -78.2% in the ES1-C13 case. Similar to the maximum temperature, the rest of the models and cases show an intermediate representation the variability of the mean temperature, with the best skills always for the cases including the ARI+ACI interactions. Results for the dust episode are shown in the right column of Fig. 9. The standard deviation tends to be overestimated by all models in the north of Africa and central Europe, and underestimated in the eastern part of the target domain. Overall, the inclusion of ARI does not lead to better skills of the models when representing the temporal variability (+2.4%), and for ARI+ACI the skill improved only marginally (reductions of -0.6%).

With respect to the minimum temperature, for the fires episode (left column of Fig. 10) all runs tend to overestimate the standard deviation. Biases of the minimum temperature standard deviation range between +0.4 K for the WRF-Chem-based simulations and +1.0 K for DE3-C11. The high-resolution CS2-C11 simulation presents the lowest bias (+0.3 K).

If considering the biases of the standard deviation, there is a slight improvement when including ARI or ARI+ACI for the fires episode, while a slight worsening is depicted for the dust case. The variations in the biases of the standard deviation are on average -2.1% and

-4.9% respectively for the ARI and ARI+ACI simulations (+3.4% and +5.4% for the dust episode).

## 3.5 Spatial variability

Taylor diagrams (Taylor, 2001) allow an easy comparison between the spatial and temporal
patterns of two fields (Rauscher et al., 2010). In Fig. 11 shows the relative spatial standard deviation (radial distance from the origin) and the correlation (the cosine of the angular coordinate) with E-Obs. Model results with good performance in terms of spatial variability and correlation are located closer to the standard deviation ratio 1 and correlation 1, which corresponds to E-OBS (indicated by the small black asterisk). For maximum, mean and
minimum temperature, the diverse models (and configurations) show a narrow spread in the representation of the spatial structure of the standard deviation.

With respect to the mean field of maximum temperature (left column in Fig. 11) all models perform well for the fires period (top row), with high spatial correlations (over 0.9) and a normalized standard deviation close to observations. However, the no radiative feedback
configuration (C11 cases in Fig. 11) represent excessive spatial variability (standard deviation ratio over 1). The spatial variability of the daily standard deviation for the ARI simulations (asterisks in Fig. 11, C12 cases), as well as for ARI+ACI simulations (squares, C13 cases) is substantially improved, despite the spatial correlation remains practically constant for all models. Since there is a positive bias in the models when representing the spatial variability
in the no radiative feedbacks simulations, the inclusion of radiative effects reduces the variability and therefore improves its spatial patterns. Analogous results can be found for the dust episode (bottom row, Fig. 11), with a larger agreement between models, and lower differences between C21, C22 and C23 cases (no feedbacks, ARI and ARI+ACI simulations, in that order).

With respect to the mean temperature (center column in Fig. 11), the models perform very similarly with each other, showing a high spatial correlation with the observations (over 0.9 for all models and cases), with a small overestimation of the spatial variability for the C11 (fire episode, no radiative feedbacks) case (top row), which improves when including

the ARI and ARI+ACI interactions. Similarly, the spatial variability is slightly overestimated for the C21 (dust, no radiative feedbacks) case, except for the DE3 model. Generally, the models better capture the spatial structure of the variability during the fires and dust cases (Fig. 11, center column) when including the radiative feedbacks. The correlation is only slightly improved for the ARI and ARI+ACI cases (except for ENS simulations, which will be discussed below), and is always higher for the mean temperature than for maximum temperature.

The minimum temperature (Fig. 11, right column) is captured with quality as the maximum and mean temperature. While for the fire episode the models (in all cases) tend to provide a higher spatial variability than the observations, the spatial variability is underestimated for the dust episode, but with a high correlation (over 0.9) for both episodes. For this variable, the improvement of including the radiative feedbacks is not so evident, since the spatial variability does not generally improve for C12, C13, C22 or C23 cases with respect to the configuration without radiative feedbacks. Moreover, the correlation coefficient is even slightly reduced with the inclusion of ARI or ARI+ACI.

Last, the added value of considering the ensemble mean of all available simulations in each episode and case is clear for the fires episode, but not that obvious for the dust period. For the fire episode, the ensemble mean outperforms individual models in terms of the standard deviation and the correlation coefficient, especially for mean temperature, where the correlation increases up to 0.99 for the ENS-C11 case. The exception is found for the ENS-C13 for minimum temperature. Generally, the skill of most models improves when aerosol-meteorology interactions are taken into account

For the dust case, the ensemble mean outperforms the individual models for representing the standard deviation (that is, the spatial variability). However, the spatial correlation coefficient is somewhat reduced as compared to the individual models.

## 4 Summary and conclusions

This study shows a collective operational evaluation of the temperature at 2 meters (maximum, mean and minimum) simulated by the coupled chemistry and meteorology models under the umbrella of COST Action ES1004 for a wildfires and a dust episode in the year
2010. The meteorological parameters considered in this assessment are important to understand the effect of the aerosol interactions with clouds and radiation. In this sense, this study complements other several analysis (e.g. Brunner et al. (2015); Forkel et al. (2015); Makar et al. (2015b)) by analyzing whether the inclusion of the radiative feedbacks improves or not the representation of the temperature field (maximum, mean and minimum)
in an ensemble of simulations.

Focusing on the bias, in both episodes there is a general underestimation of the studied variables, being most noticeable in maximum temperature. In general, there is not a straightforward conclusion with respect to the improvement (or not) of the bias when introducing aerosol radiative feedbacks. Broadly, the biases are improved when including ARI or
ARI+ACI in the dust case, but no evident improvements are found for the heatwave/wildfires episode. Although the ensemble does not outperform the individual models (in general), the improvements found when including ARI and ARI+ACI are by far more remarkable for the ensemble than for the individual models.

With respect to the temporal correlation, maximum and mean temperatures in the fires
and dust episode show higher correlations over most of the domain when considering C11 case with respect to the E-OBS database than minimum temperature. During these episodes, a twofold conclusion can be obtained: (1) the ensemble of simulations always outperforms the representation of the temporal variability of the series; and (2) an improvement of the $\rho^2$ coefficient is found when considering ARI or ARI+ACI feedbacks (in both
episodes).

Regarding the temporal variability, during the fire episode there is a general pronounced overestimation of the standard deviation of the studied variables. Here, the inclusion of aerosol feedbacks largely improves the representation of the temporal variability of the three

studied variables (reduction of the bias of the standard deviation) showing the best skills for the cases including the ARI+ACI interactions, with a reduction of bias of the standard deviation by as much as 75%. Very similar results can be found for the dust episode. Generally, it is for the temporal variability where the inclusion of the aerosol radiative feedbacks shows the largest improvements and results in an added value of the computational effort made to include direct aerosol radiation interactions and aerosol cloud interactions in the models. Last, with respect to the spatial variability for maximum and mean temperature, the inclusion of radiative effects reduces the variability and improves the spatial patterns for both episodes. For the minimum temperature, the improvement of including the radiative feedbacks is less evident.

In order to further investigate the impact of including the aerosol interactions in online coupled models, more episodes with effects on the aerosol-radiation-cloud interactions should be considered. In this work, the fires episode represents a situation of clear skies, and therefore the ARI feedbacks are dominant. The dust episode election permits to study aerosol-cloud interaction, most of the ARI+ACI differences found in the models with respect to the base case were found over the Mediterranean sea. Since the observational data E-OBS only has values over land, the effect of ARI+ACI were not evaluation here. Unfortunately part of the interpretation of the results may be missed due to the unavailability of this database over the ocean. Furthermore, it should be pointed out that all results for the ARI+ACI cases were from WRF-Chem simulations, which may bias the ARI+ACI results towards the behaviour of this model.

There are still modelling issues regarding the representation of the field of temperature, where maximum temperatures are underestimated and minimum temperatures are overestimated and the inclusion of the aerosol feedbacks does not improve this situation. Nevertheless, in this study, a general improvement of the temporal variability and correlation has been seen. These improvements may be important not only for certain episodes, as analyzed here, by also for the representation of the climatology of temperatures. However, climatic-representative periods should be covered in further studies.

*Acknowledgements.* We acknowledge the E-OBS dataset from the EU-FP6 project ENSEMBLES (http://ensembles-eu.metoffice.com) and the data providers in the ECAD project (http://www.ecad.eu). This research have been carried out under the support of the Project REPAIR (CGL2014-59677-R), funded by the Spanish Ministerio de Economía y Competitividad (MINECO) and the FEDER Euro-
pean program. Rocío Baró acknowledges the FPU scholarship (Ref. FPU12/05642) by the Spanish Ministerio de Educación, Cultura y Deporte. Pedro Jiménez-Guerrero acknowledges the fellowship 19677/EE/14 of the Programme Jiménez de la Espada de Movilidad, Cooperación e Internacionalización (Fundación Séneca-Agencia de Ciencia y Tecnología de la Región de Murcia, PCTIRM 2011-2014). Finally, we acknowledge the support of the European groups through COST Action
ES1004 EuMetChem and the Air Quality Modelling Evaluation International Initiative (AQMEII) and the EuMetChem COST ACTION ES1004 have supported these works.

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

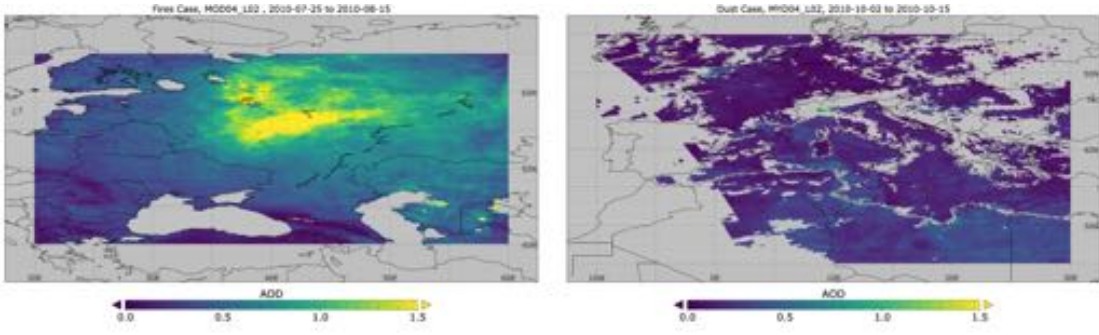

**Figure 1.** (Top row) Aerosol Optical Depth (AOD) at 550 nm for the fires (left) and dust (right) episodes, as derived from MODIS. The panel below represents the bias for the fires (left) and dust (right) episodes of each simulation with respect to the MODIS AOD. NRF: no radiative feedbacks; ARI: aerosol-radiation interactions; ARI+ACI: as ARI including aerosol-cloud interactions.

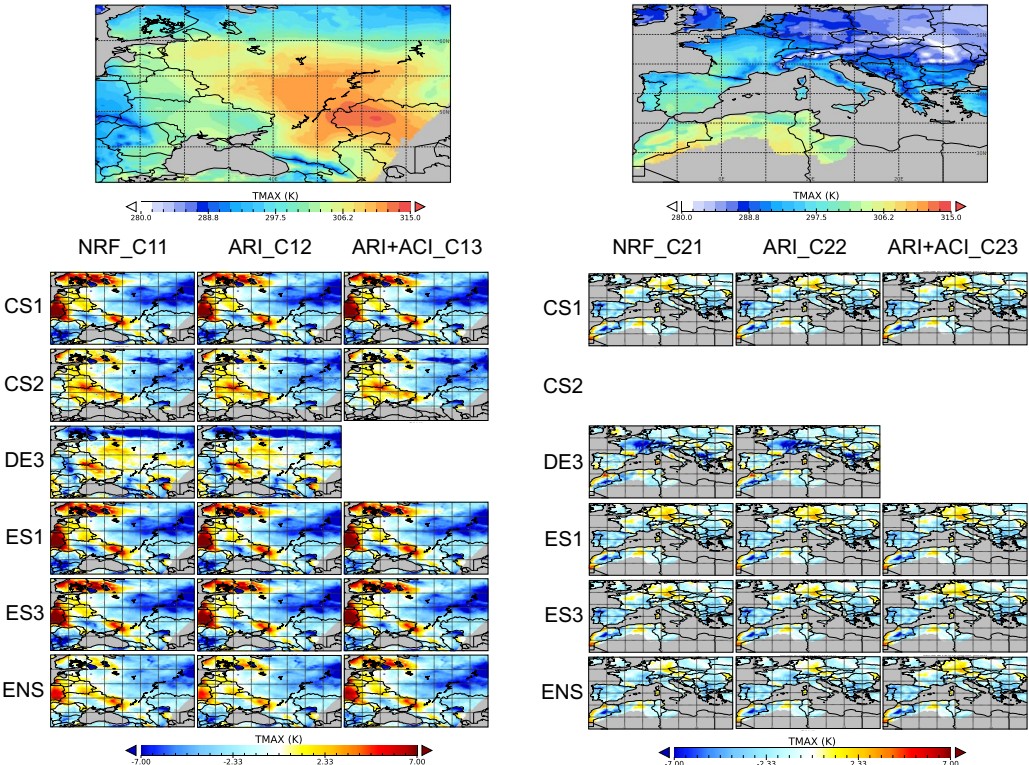

**Figure 2.** (Top row) Maximum temperature (TMAX) for the fires (left) and dust (right) episodes, as derived from E-OBS database (in K). The panel below represents the bias for the fires (left) and dust (right) episodes of each simulation with respect to the E-OBS database. NRF: no radiative feedbacks; ARI: aerosol-radiation interactions; ARI+ACI: as ARI including aerosol-cloud interactions.

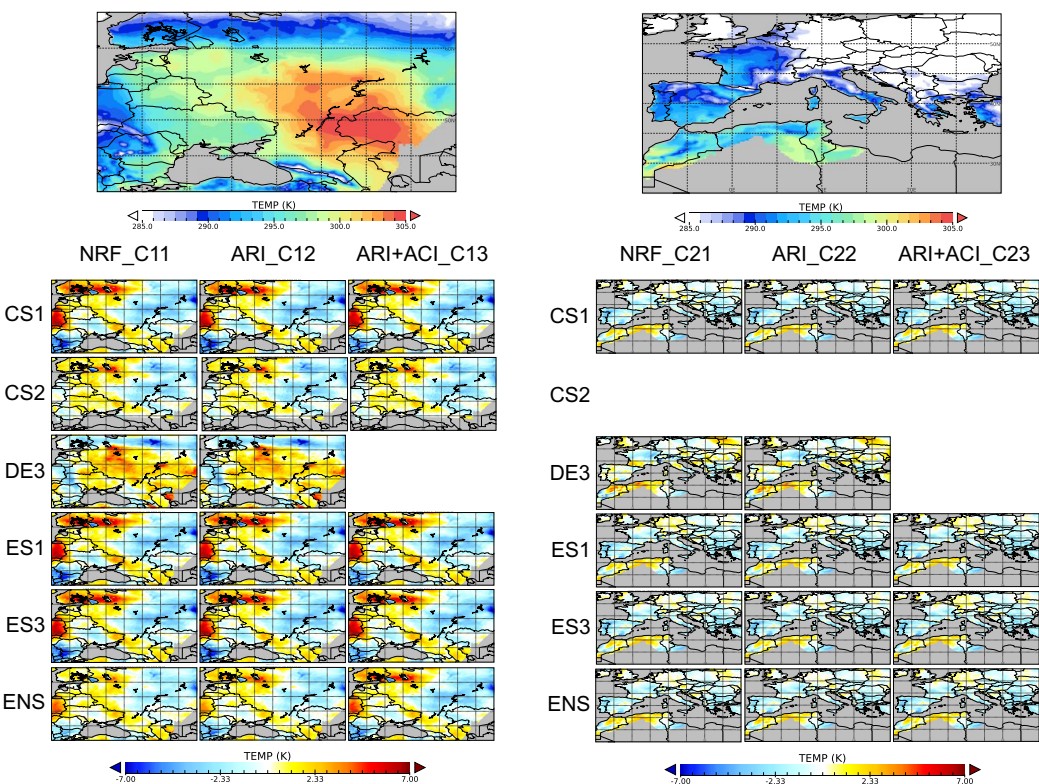

**Figure 3.** (Top row) Mean temperature (TEMP) for the fires (left) and dust (right) episodes, as derived from E-OBS database (in K). The panel below represents the bias for the fires (left) and dust (right) episodes of each simulation with respect to the E-OBS database. NRF: no radiative feedbacks; ARI: aerosol-radiation interactions; ARI+ACI: as ARI including aerosol-cloud interactions.

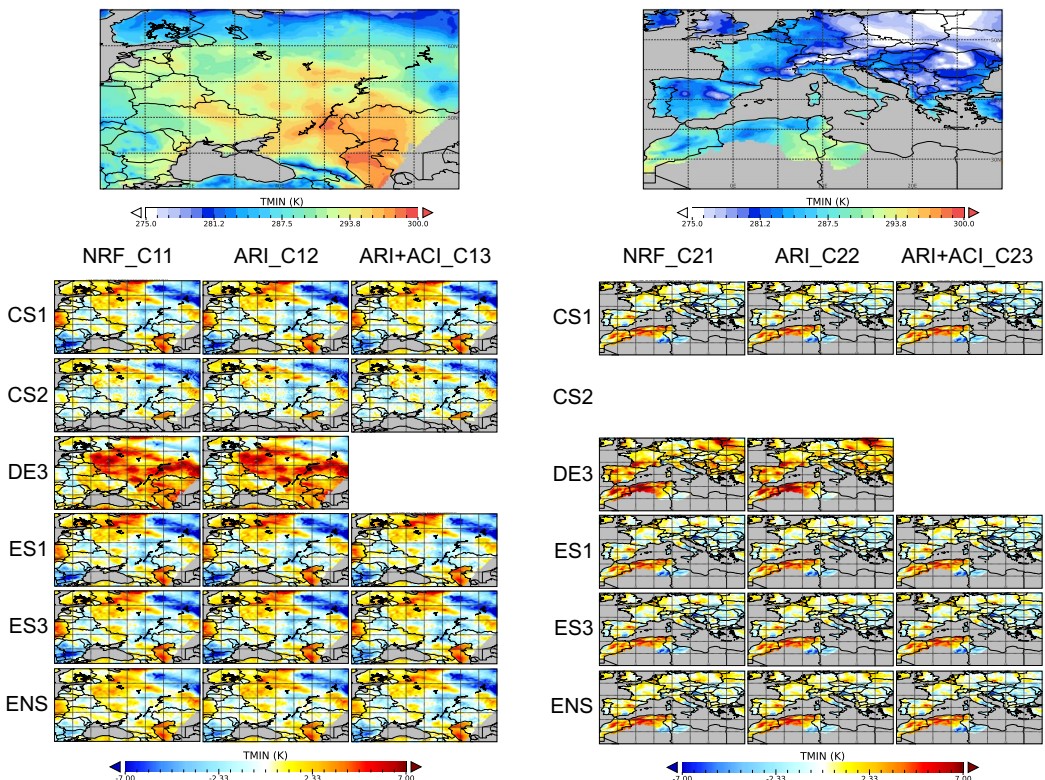

**Figure 4.** (Top row) Minimum temperature (TMIN) for the fires (left) and dust (right) episodes, as derived from E-OBS database (in K). The panel below represents the bias for the fires (left) and dust (right) episodes of each simulation with respect to the E-OBS database. NRF: no radiative feedbacks; ARI: aerosol-radiation interactions; ARI+ACI: as ARI including aerosol-cloud interactions.

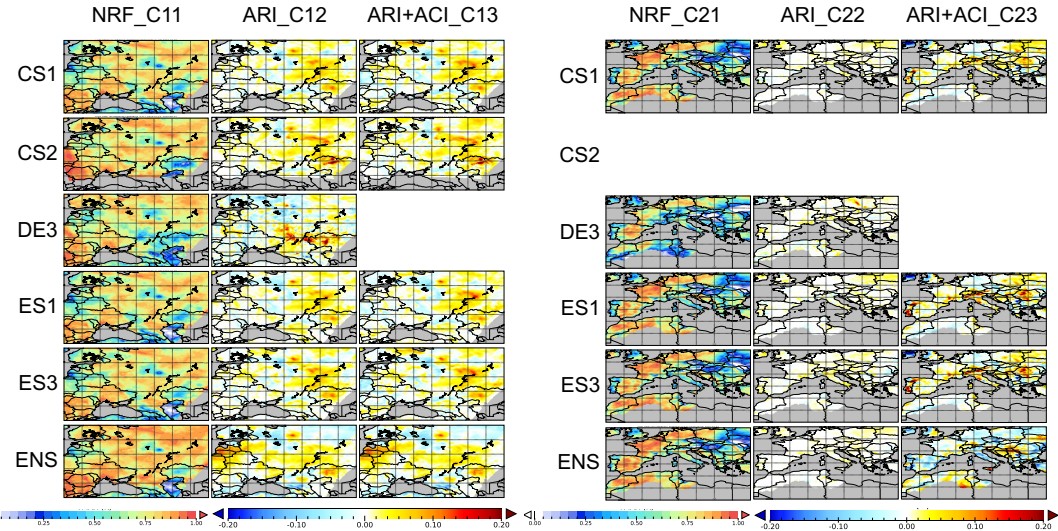

**Figure 5.** (Top row) Time determination coefficient ($\rho^2$) (model vs. E-OBS) of the maximum temperature (TMAX) for the fires (left panel) and dust (right panel) episodes. The first column in each panel below represents the value of $\rho^2$ of the no radiative feedback case with respect to the E-OBS database. The center and right columns indicate the increase (red values) or decrease (blue value) of each simulation with respect to the case not including feedbacks. NRF: no radiative feedbacks; ARI: aerosol-radiation interactions; ARI+ACI: as ARI including aerosol-cloud interactions.

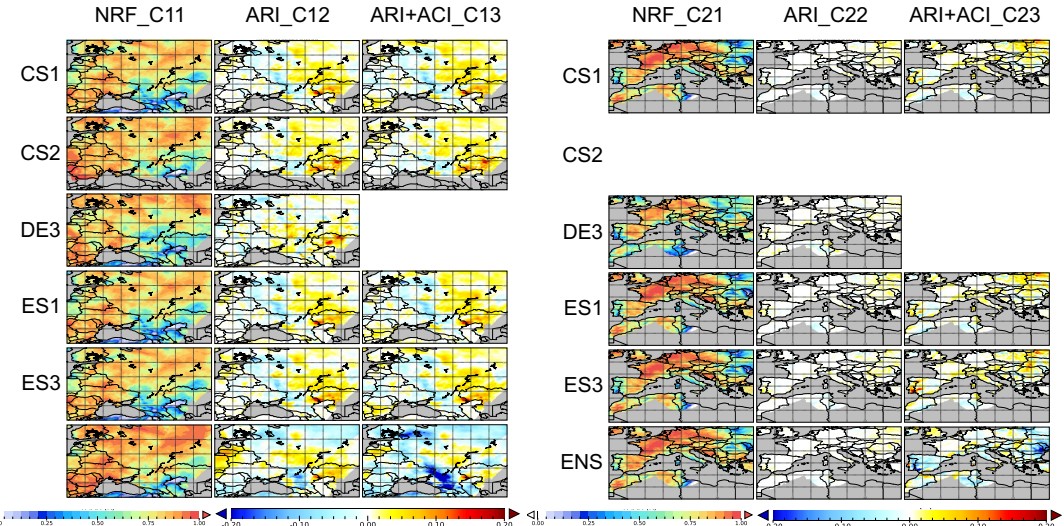

**Figure 6.** (Top row) Time determination coefficient ($\rho^2$) (model vs. E-OBS) of the mean temperature (TEMP) for the fires (left panel) and dust (right panel) episodes. The first column in each panel below represents the value of $\rho^2$ of the no radiative feedback case with respect to the E-OBS database. The center and right columns indicate the increase (red values) or decrease (blue value) of each simulation with respect to the case not including feedbacks. NRF: no radiative feedbacks; ARI: aerosol-radiation interactions; ARI+ACI: as ARI including aerosol-cloud interactions.

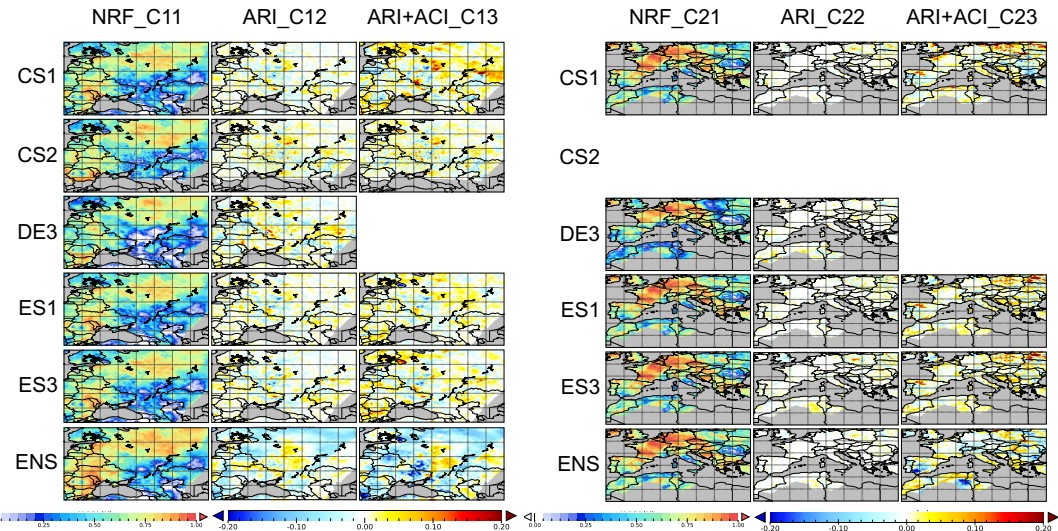

**Figure 7.** (Top row) Time determination coefficient ($\rho^2$) (model vs. E-OBS) of the minimum temperature (TMIN) for the fires (left panel) and dust (right panel) episodes. The first column in each panel below represents the value of $\rho^2$ of the no radiative feedback case with respect to the E-OBS database. The center and right columns indicate the increase (red values) or decrease (blue value) of each simulation with respect to the case not including feedbacks. NRF: no radiative feedbacks; ARI: aerosol-radiation interactions; ARI+ACI: as ARI including aerosol-cloud interactions.

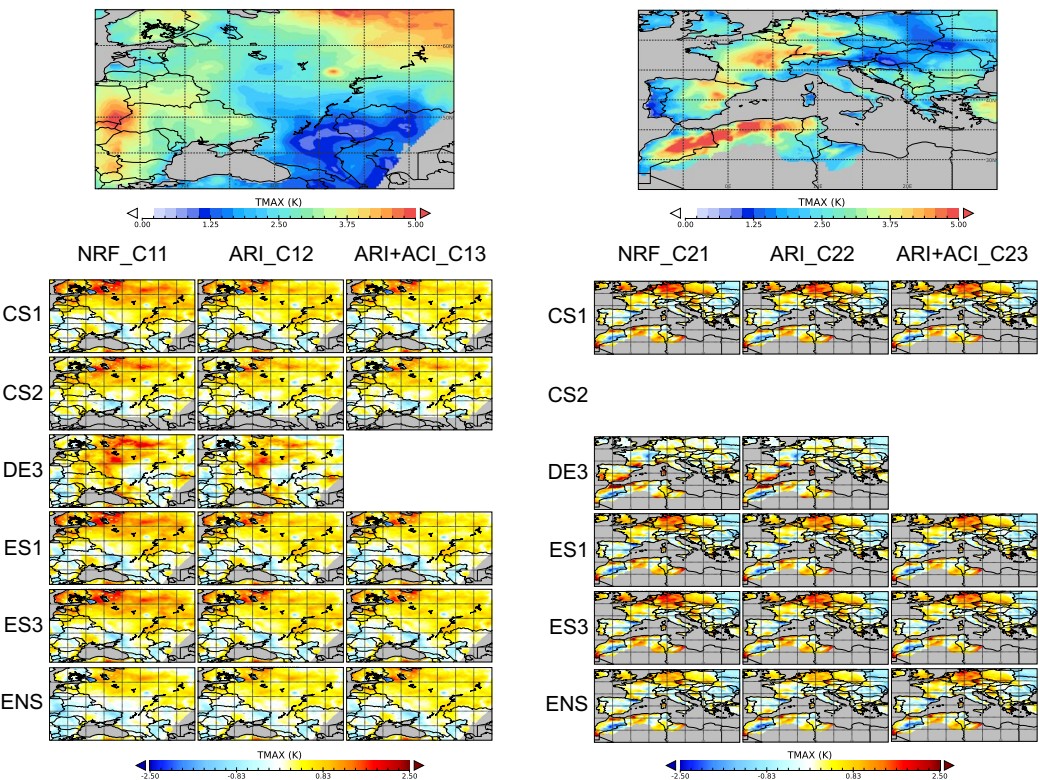

**Figure 8.** (Top row) Standard deviation (STD) of the maximum temperature (TMAX) for the fires (left) and dust (right) episodes, as derived from E-OBS database (in K). The panel below represents the bias for the standard deviation of the fires (left) and dust (right) episodes of each simulation with respect to the E-OBS database. NRF: no radiative feedbacks; ARI: aerosol-radiation interactions; ARI+ACI: as ARI including aerosol-cloud interactions.

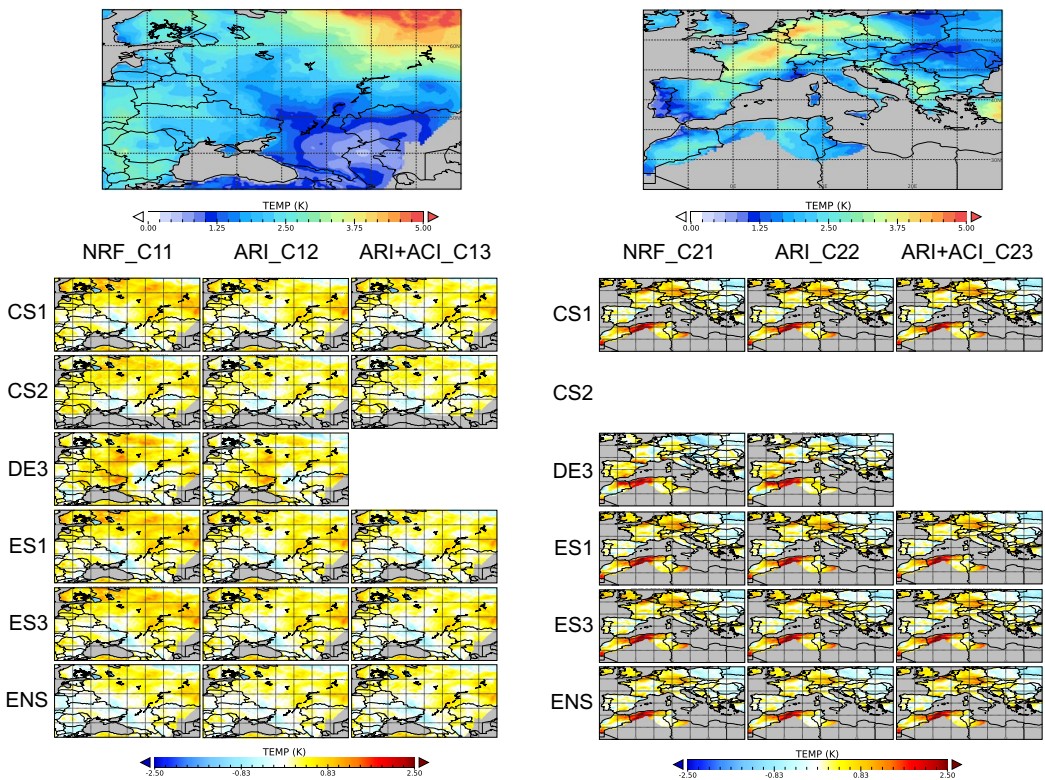

**Figure 9.** (Top row) Standard deviation (STD) of the mean temperature (TEMP) for the fires (left) and dust (right) episodes, as derived from E-OBS database (in K). The panel below represents the bias for the standard deviation of the fires (left) and dust (right) episodes of each simulation with respect to the E-OBS database. NRF: no radiative feedbacks; ARI: aerosol-radiation interactions; ARI+ACI: as ARI including aerosol-cloud interactions.

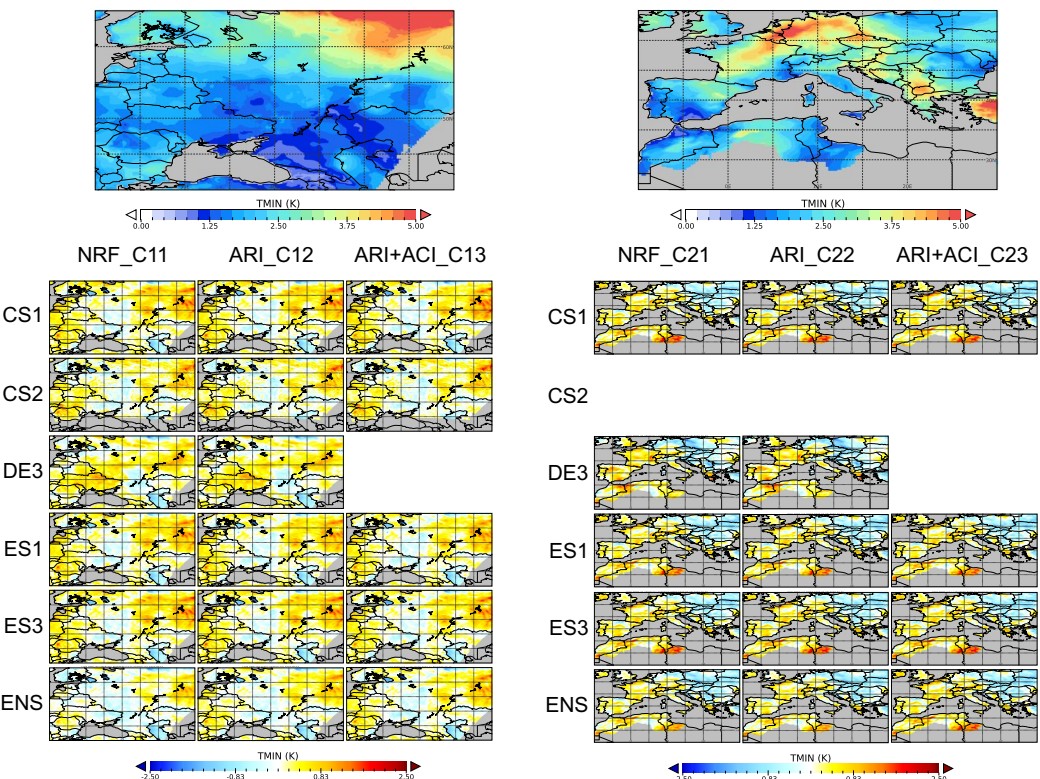

**Figure 10.** (Top row) Standard deviation (STD) of the minimum temperature (TMIN) for the fires (left) and dust (right) episodes, as derived from E-OBS database (in K). The panel below represents the bias for the standard deviation of the fires (left) and dust (right) episodes of each simulation with respect to the E-OBS database. NRF: no radiative feedbacks; ARI: aerosol-radiation interactions; ARI+ACI: as ARI including aerosol-cloud interactions.

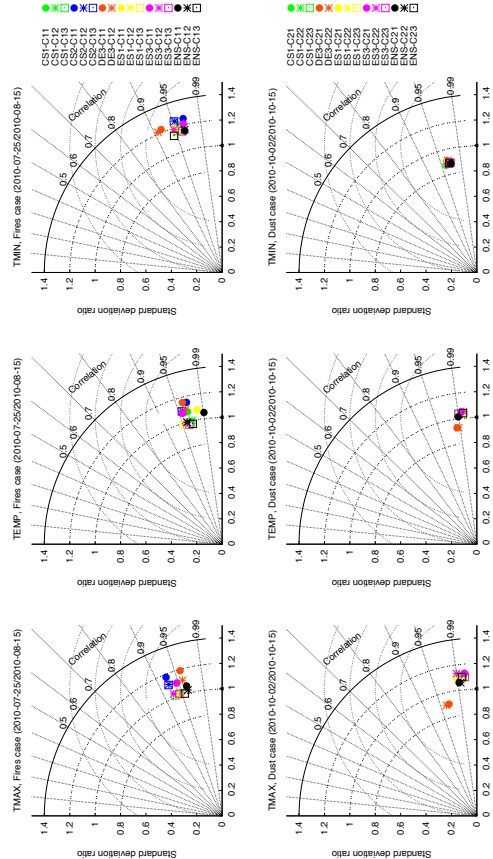

**Figure 11.** Taylor diagrams for (left) maximum temperature, (center) mean temperature, and (right) minimum temperature for the simulations included in the analysis. The top row represents the Taylor diagrams for the fires episode, while the bottom row stands for the dust episode. The cases included are: no radiative feedbacks (filled circle), ARI (asterisk) and ARI+ACI (empty squares). Each configuration is shown in a different color: CS1 (green), CS2 (dark blue), DE3 (red), ES1 (yellow), ES3 (pink) and ENS (black).

**Table 1.** Modelling systems participating and their contributions to the case studies.

| | CS1 | CS2 | DE3 | ES1 | ES3 |
|---|---|---|---|---|---|
| Lead Institution | UL, KIT/IMK-IFU* | UL, KIT/IMK-IFU* | IFT Leipzig | U. Murcia | UPM-ESMG |
| Model | WRF-Chem | WRF-Chem | COSMO-MUSCAT | WRF-Chem | WRF-Chem |
| Episode | Fire, Dust | Fire | Fire, Dust | Fire, Dust | Fire, Dust |
| Runs | NRF, ARI, ARI+ACI | NRF, ARI, ARI+ACI | NRF, ARI | NRF, ARI, ARI+ACI | NRF, ARI, ARI+ACI |
| Resolution | 23 km | 9.9 km | 0.125 deg. | 23 km | 23 km |
| Microphysics | Morrison | Morrison | Kessler-type bulk | Lin | Morrison |
| SW Radiation | RRTMG | RRTMG | $\delta$-2-stream | RRTMG | RRTMG |
| LW Radiation | RRTMG | RRTMG | $\delta$-2-stream | RRTMG | RRTMG |
| PBL/turbulence | YSU | YSU | Prognostic TKE | YSU | YSU |
| Biogenic model | MEGAN (Guenther et al., 2006) | MEGAN | Guenther et al. (1993) | MEGAN | MEGAN |
| Gas phase | RADM2 modified | RADM2 modified | RACM-MIM2 | RADM2 | CBMZ |
| Aerosol | MADE / SORGAM | MADE / SORGAM | Simpson et al. (2003) | MADE/SORGAM | MOSAIC 4 bins |
| Model reference | Grell et al. (2005); Forkel et al. (2015) | Grell et al. (2005); Forkel et al. (2015) | Wolke et al. (2012) | Grell et al. (2005) | Grell et al. (2005) |

*Joint effort, also including ZAMG, RSE, UPM-ESMG

**Table 2.** Domain-averaged bias (in K) for the fires (C1X) and dust (C2X) episodes of each simulation with respect to the E-OBS database. NRF: no radiative feedbacks; ARI: aerosol-radiation interactions; ARI+ACI: as ARI including aerosol-cloud interactions.

| Bias TMAX | CS1 | CS2 | DE3 | ES1 | ES3 | ENS |
|---|---|---|---|---|---|---|
| C11(NRF) | -2.140 | -2.120 | -1.164 | -2.047 | -2.141 | -1.945 |
| C12(ARI) | -2.424 | -2.376 | -1.566 | -2.325 | -2.408 | -2.242 |
| C13(ARI+ACI) | -2.397 | -2.376 | | -2.265 | -2.336 | -2.387 |
| C21(NRF) | -0.854 | | -1.006 | -0.564 | -0.852 | -0.820 |
| C22(ARI) | -0.950 | | -1.039 | -0.636 | -0.967 | -0.898 |
| C23(ARI+ACI) | -0.816 | | | -0.646 | -0.755 | -0.739 |
| Bias TEMP | CS1 | CS2 | DE3 | ES1 | ES3 | ENS |
| C11(NRF) | -0.460 | -0.455 | 0.992 | -0.409 | -0.459 | -0.187 |
| C12(ARI) | -0.745 | -0.720 | 0.721 | -0.696 | -0.767 | -0.443 |
| C13(ARI+ACI) | -0.724 | -0.715 | | -0.642 | -0.703 | -0.376 |
| C21(NRF) | 0.359 | | 0.790 | 0.446 | 0.358 | 0.487 |
| C22(ARI) | 0.289 | | 0.771 | 0.390 | 0.289 | 0.443 |
| C23(ARI+ACI) | 0.339 | | | 0.384 | 0.383 | 0.368 |
| Bias TMIN | CS1 | CS2 | DE3 | ES1 | ES3 | ENS |
| C11(NRF) | -0.019 | -0.157 | 2.680 | -0.034 | -0.019 | 0.484 |
| C12(ARI) | -0.032 | -0.211 | 2.640 | -0.044 | -0.035 | 0.485 |
| C13(ARI+ACI) | -0.040 | -0.212 | | -0.050 | -0.040 | -0.047 |
| C21(NRF) | 0.596 | | 1.792 | 0.526 | 0.595 | 0.876 |
| C22(ARI) | 0.581 | | 1.791 | 0.390 | 0.515 | 0.616 |
| C23(ARI+ACI) | 0.509 | | | 0.516 | 0.604 | 0.541 |

**Table 3.** Bias (in K) in those areas affected by high aerosol optical depths (1-hr AOD>1.0 for the fires, C1X case, and AOD>0.5 for the dust, C2X case) with respect to the E-OBS database. NRF: no radiative feedbacks; ARI: aerosol-radiation interactions; ARI+ACI: as ARI including aerosol-cloud interactions.

| Bias TMAX | CS1 | CS2 | DE3 | ES1 | ES3 | ENS |
|---|---|---|---|---|---|---|
| C11(NRF) | -4.616 | -5.301 | -2.970 | -4.406 | -4.870 | -4.427 |
| C12(ARI) | -5.073 | -5.742 | -3.541 | -4.926 | -5.315 | -4.892 |
| C13(ARI+ACI) | -5.051 | -5.735 | | -4.874 | -5.233 | -5.329 |
| C21(NRF) | -3.382 | | -3.968 | -2.814 | -3.256 | -3.501 |
| C22(ARI) | -3.486 | | -4.012 | -2.924 | -3.346 | -3.584 |
| C23(ARI+ACI) | -3.334 | | | -2.728 | -3.001 | -2.504 |
| Bias TEMP | CS1 | CS2 | DE3 | ES1 | ES3 | ENS |
| C11(NRF) | -0.405 | -0.738 | 1.008 | -0.308 | -0.316 | -0.090 |
| C12(ARI) | -0.818 | -1.132 | 0.612 | -0.774 | -0.737 | -0.485 |
| C13(ARI+ACI) | -0.798 | -1.115 | | -0.709 | -0.670 | -0.841 |
| C21(NRF) | -0.429 | | -0.200 | -0.080 | -0.534 | -0.451 |
| C22(ARI) | -0.503 | | -0.237 | -0.165 | -0.592 | -0.509 |
| C23(ARI+ACI) | -0.404 | | | -0.155 | -0.496 | 0.080 |
| Bias TMIN | CS1 | CS2 | DE3 | ES1 | ES3 | ENS |
| C11(NRF) | 3.171 | 3.313 | 4.703 | 3.122 | 3.666 | 3.742 |
| C12(ARI) | 2.996 | 3.125 | 4.566 | 2.926 | 3.439 | 3.568 |
| C13(ARI+ACI) | 2.980 | 3.122 | | 2.914 | 3.414 | 3.244 |
| C21(NRF) | 2.394 | | 3.476 | 2.576 | 2.067 | 2.491 |
| C22(ARI) | 2.348 | | 3.450 | 2.516 | 2.051 | 2.461 |
| C23(ARI+ACI) | 2.323 | | | 2.508 | 1.962 | 2.175 |

**Table 4.** Domain-averaged coefficient of determination ($\rho^2$) for the fires (C1X) and dust (C2X) episodes of each simulation with respect to the E-OBS database. NRF: no radiative feedbacks; ARI: aerosol-radiation interactions; ARI+ACI: as ARI including aerosol-cloud interactions.

| $\rho^2$ TMAX | CS1 | CS2 | DE3 | ES1 | ES3 | ENS |
|---|---|---|---|---|---|---|
| C11(NRF) | 0.658 | 0.697 | 0.713 | 0.648 | 0.658 | 0.753 |
| C12(ARI) | 0.670 | 0.710 | 0.714 | 0.660 | 0.671 | 0.748 |
| C13(ARI+ACI) | 0.670 | 0.710 | | 0.658 | 0.669 | 0.719 |
| C21(NRF) | 0.857 | | 0.757 | 0.861 | 0.857 | 0.870 |
| C22(ARI) | 0.859 | | 0.759 | 0.863 | 0.859 | 0.871 |
| C23(ARI+ACI) | 0.860 | | | 0.862 | 0.861 | 0.864 |
| $\rho^2$ TEMP | CS1 | CS2 | DE3 | ES1 | ES3 | ENS |
| C11(NRF) | 0.757 | 0.775 | 0.785 | 0.753 | 0.757 | 0.843 |
| C12(ARI) | 0.760 | 0.781 | 0.790 | 0.755 | 0.759 | 0.831 |
| C13(ARI+ACI) | 0.761 | 0.781 | | 0.757 | 0.761 | 0.818 |
| C21(NRF) | 0.893 | | 0.836 | 0.896 | 0.893 | 0.900 |
| C22(ARI) | 0.894 | | 0.837 | 0.897 | 0.894 | 0.902 |
| C23(ARI+ACI) | 0.895 | | | 0.897 | 0.895 | 0.897 |
| $\rho^2$ TMIN | CS1 | CS2 | DE3 | ES1 | ES3 | ENS |
| C11(NRF) | 0.519 | 0.525 | 0.554 | 0.516 | 0.519 | 0.614 |
| C12(ARI) | 0.514 | 0.520 | 0.555 | 0.513 | 0.511 | 0.586 |
| C13(ARI+ACI) | 0.520 | 0.521 | | 0.514 | 0.519 | 0.586 |
| C21(NRF) | 0.816 | | 0.764 | 0.821 | 0.816 | 0.832 |
| C22(ARI) | 0.816 | | 0.764 | 0.820 | 0.817 | 0.831 |
| C23(ARI+ACI) | 0.819 | | | 0.820 | 0.818 | 0.832 |

**Table 5.** Coefficient of determination ($\rho^2$) in those areas affected by high aerosol optical depths (1-hr AOD>1.0 for the fires, C1X case, and AOD>0.5 for the dust, C2X case) with respect to the E-OBS database. NRF: no radiative feedbacks; ARI: aerosol-radiation interactions; ARI+ACI: as ARI including aerosol-cloud interactions.

| $\rho^2$ TMAX | CS1 | CS2 | DE3 | ES1 | ES3 | ENS |
|---|---|---|---|---|---|---|
| C11(NRF) | 0.614 | 0.579 | 0.743 | 0.552 | 0.552 | 0.781 |
| C12(ARI) | 0.616 | 0.587 | 0.760 | 0.556 | 0.556 | 0.788 |
| C13(ARI+ACI) | 0.618 | 0.588 | | 0.560 | 0.560 | 0.790 |
| C21(NRF) | 0.849 | | 0.848 | 0.824 | 0.823 | 0.876 |
| C22(ARI) | 0.853 | | 0.851 | 0.827 | 0.827 | 0.878 |
| C23(ARI+ACI) | 0.856 | | | 0.832 | 0.833 | 0.886 |

| $\rho^2$ TEMP | CS1 | CS2 | DE3 | ES1 | ES3 | ENS |
|---|---|---|---|---|---|---|
| C11(NRF) | 0.656 | 0.609 | 0.800 | 0.594 | 0.729 | 0.814 |
| C12(ARI) | 0.656 | 0.609 | 0.811 | 0.596 | 0.734 | 0.816 |
| C13(ARI+ACI) | 0.656 | 0.609 | | 0.608 | 0.736 | 0.816 |
| C21(NRF) | 0.901 | | 0.897 | 0.863 | 0.873 | 0.921 |
| C22(ARI) | 0.903 | | 0.899 | 0.866 | 0.875 | 0.924 |
| C23(ARI+ACI) | 0.904 | | | 0.870 | 0.885 | 0.925 |

| $\rho^2$ TMIN | CS1 | CS2 | DE3 | ES1 | ES3 | ENS |
|---|---|---|---|---|---|---|
| C11(NRF) | 0.462 | 0.403 | 0.622 | 0.414 | 0.510 | 0.632 |
| C12(ARI) | 0.464 | 0.400 | 0.630 | 0.419 | 0.507 | 0.634 |
| C13(ARI+ACI) | 0.465 | 0.400 | | 0.419 | 0.507 | 0.577 |
| C21(NRF) | 0.833 | | 0.835 | 0.797 | 0.867 | 0.877 |
| C22(ARI) | 0.836 | | 0.837 | 0.798 | 0.872 | 0.880 |
| C23(ARI+ACI) | 0.836 | | | 0.799 | 0.886 | 0.870 |