# Peer review of "Regional effects of atmospheric aerosols on temperature: an evaluation of an ensemble of on-line coupled models"

_Atmospheric Chemistry and Physics, 2016_

## Referee Comment (RC1) · Anonymous Referee #1 · 25 Jan 2017

General comments The article has a very clear objective of evaluating the simulations of coupled models including aerosol interactions with radiation and clouds with respect to surface temperatures. This is clearly of general interest since it is an impact on weather forecasting that may motivate changes in operational models around the world. The improvement of temporal variability is an important result.

The paper is written like a report. There is very little discussion on why given models may perform better than others. There is also a lack of discussion on how well the aerosol concentrations for the two episodes compare with actual observations. The reader is left without means to judge whether this is a convincing case or not. In the conclusions, the authors reinforce this feeling by saying that this evaluation should be

performed for cases with "episodes with stronger effects on the aerosol cloud interactions" and mentioning that in one of the cases larger concentration were found over the Mediterranean Sea where the evaluation in not performed.

The paper also lacks objective definition of some parameters and procedures. The numbers that summarize results should be organized in tables so they can be easily compared.

Specific comments

Page 4, lines 20-23 – please correct sentence structure

Page 35 – Table 1 – define CS1, CS2, DE3, ES1, ES3, in the text you use things like, C11, C12...etc, this should be defined in the text.

Page 6, line 8 you mention annual emissions, what is actually used in a daily/hourly basis? Are these cases significant from the point of view of high emissions and concentration of aerosol over land?

Page 6, line 17. The reference Im et al (2015b) states that the PM values show large underestimations, particularly because of dust and sea salt emissions. In the cases used in this paper, how do this values compare to station data?

Pages 7, Equation 2, objectively define ˆ ; what is the operation defined in eq. 2? What is Vick ?

Page 9, lines 3-5, for the whole period, 60 days in one case and 30 days in the other case? Only in figure 10 the exact period is mentioned. Please state that in the text.

Page 9, lines 6-28 the average numbers of the bias for each case and run should be summarized in a Table.

Page 9 – a figure with the average concentrations of PM should be included to help the discussion on the bias. Visual inspection of figures 1 – 3 shows very similar results and does not help the case of the aerosol effect on temperature. What is needed is

a way to represent the effect on temperature in places where there is a high concentration of aerosol. You could choose a given simulation time with very high aerosol concentrations and show model performance for temperatures.

Page 10-11, the average numbers of the coefficient of determination and improvement or not on standard deviation in the analysis of temporal variability for each case and run should be summarized in a Table.

Page 14, lines 23-26 – for Tmin the case ENS-C13 is perhaps an exception?

---

## Referee Comment (RC2) · Anonymous Referee #2 · 4 Feb 2017

The manuscripts analyses the effects of aerosol-cloud-radiation interactions on temperature for two study cases over Europe (smoke and dust episodes) using an ensemble of models and surface observations. The results points towards an improvement of spatial and temporal correlations when adding these interactions. The paper is well written and the topic is in the scope of the Journal. I have some comments and suggestions below.

General comments

The authors jump directly to the statistical assessment but it would be good to also provide a figure(s) showing observed AOD and clouds for the period of analysis and then how each model was able to represent them. If this is performed in other papers

please include it anyways for context.

In general whether aerosols effects improve the simulation depends on the initial bias of the model. However, since the authors perform their statistical analysis over the whole domain, the overlaying bias in regions with little aerosol impact could be driving the results in the wrong direction, especially as the biases are very inhomogeneous throughout the domain. I suggest the authors to restrict their analysis only to regions where large aerosol impact is expected. For instance, bias could be computed only for regions where AOD is over a fixed threshold, or you could weigh the bias by AOD. If the former was done then the regions not included in the analysis could be shaded in the figures showing the different statistics.

Given that the radiation effects tend to be maximized in the early morning and later afternoon (light path through the atmosphere is longer) previous studies have used other metrics to assess the impacts of aerosols. For instance, the trend in daytime temperature range (See Xing at al., 2015). This metric and others could also be included in this study.

The Results section tends to focus on presenting results and not providing much explanation why the findings happen. This would be fine if there was a discussion section, but this is not the case. I encourage the authors to add more of these details to the Results sections. Some examples can be found below.

Comments by line

Page 2, line 10. Somewhere in the text define "EuMetChem COST Action ES1004"

Page 2, Line 19. I think that the statement "especially for those areas closest to emissions sources of atmospheric aerosols" is not explained or mentioned in the text.

Page 9, lines 14-15. Can you elaborate more why it is remarkable? I don't see much change from the figures. It would be good if you added a % decrease of the bias as for ARI.

Interactive
comment

Section 3.2. What strikes me the most from these results are the differences between the smoke and the dust case. For the smoke case, most of the changes come from the radiation effects, while for the dust effects most changes come from aerosol-cloud effects. Can you explain why this happens? Does the dust case has more clouds over the domain? Is it related to better cirrus representation?

Section 3.3. Why some of the models show large variations and other just intermediate? Can you elaborate?

Figure 10. Because of the narrow spread for all cases, it would be nice to plot a zoomed-in version of the taylor plot to better read the differences between the simulations.

Technical corrections

Page 4, line 22. Erase "are run"

Equation 2. There is either an error on the notation or a variable is not defined ($V\hat{}k\_ic$ )

Page 15, line 11. Fix the forkeletal2015 reference

References

Xing, J., R. Mathur, J. Pleim, C. Hogrefe, C.-M. Gan, D. C. Wong, C. Wei, and J. Wang (2015), Air pollution and climate response to aerosol direct radiative effects: A modeling study of decadal trends across the northern hemisphere, J. Geophys. Res. Atmos., 120, 12,221–12,236, doi:10.1002/2015JD023933.

---

## Referee Comment (RC3) · Anonymous Referee #3 · 6 Feb 2017

**Regional effects of atmospheric aerosols on temperature:**
**an evaluation of an ensemble of on-line coupled models**

**for Atmospheric Chemistry and Dynamics, February 2017**

This paper describes the results of a model intercomparison study involving a handful of models comparing the impact of including aerosol direct and indirect effects on surface temperature. The paper is myopic, as it only describes the comparison of simulated surface temperatures against a single gridded temperature data set for daily mean, minimum, and maximum temperatures. No attempt is made to provide any attribution for the identified differences in model behaviors. As such, the paper is simplistic and does not add a lot to the published base of research. The paper basically shows that including aerosol effects generally leads to a small improvement in the simulated surface temperature. Whether this is sufficient for justifying publication in ACP will need to be determined by the editor.

Grammatically, the paper is fairly well written with only a handful of necessary corrections in this area. I found about a dozen issues of extra words, bad commas, and the like. As the reviewer's task is to focus primarily on the science, fixing these issues will be left to a copy editor who can be more thorough.

**Major Comments**

The paper could be greatly strengthened by looking at why the model results improve, or at least by providing additional information to help readers gain context. This could be done by looking at the energy budgets. The benefit of having the range of models in the intercomparison is that one can examine if the aerosol-related improvements occur for the same reason in each model, or if there are compensating effects that lead to interesting nonlinearities.

Another way the paper could be improved is by looking beyond daily values. Only looking at daily values hides a lot of model deficiencies. Comparing the models against hourly temperature data, as well as moisture and PM2.5 amounts, would provide much more detail for understanding why the models change when including aerosol feedbacks. This would also bring the plume behavior of local aerosol sources more into play.

Gridded temperature data sets typically show a range of uncertainties due to different methodologies used to spatially distribute and average point observations. What is the uncertainty of the E-OBS data set? How does it compare to other gridded data sets available for the region? How does it compare to analyses, such as from the IFS model, that incorporate the observations with data assimilation? Most likely, the differences identified between running the models with and without aerosol feedbacks is smaller than the differences between observation data sets.

The figures with maps are presented in such a way that one cannot gain a clear quantitative understanding of how the models differ beyond very large differences. Readers are asked to look for differences on the order of tenths or hundredths of a degree in a color scale ranging across 5 K while the maps are essentially postage stamp size. In most cases, the maps look identical without extremely close examination. A better way needs to be found to present this information. One or two figures like this can be used to make the overall point and give the spatial structure of the typical bias. However, nine of these figures becomes tedious to read and they end up not conveying the intended information.

Top of p. 5: The "third" configuration needs to be defined in relation to the handling of aerosol and/or cloud droplet assumptions when aerosol-cloud interactions are disabled. Otherwise, the results are just a sensitivity test of a particular model that are not comparable to other models. The results and any subsequent conclusions are dependent upon how the models are tuned when the interactions are disabled. One needs to define an appropriate scenario for the comparison. This differs from comparisons of configurations 1 and 2 for the aerosol-radiation interactions because it is possible to run a model without aerosol impacting radiation and still get a physically reasonable result. However, one cannot run a model without aerosol and still form clouds, since the aerosols are required for forming cloud droplets in almost all physically relevant conditions. So, even "without" aerosol-cloud interactions there are still significant assumptions built into the models to account for the ACI processes.

**Minor Comments**

p. 4, l. 27: "Three different cases" is better phrased as "Three different configurations." "Cases" implies different dates and "configurations" is more specific to what is being described.

p. 5, l. 10: The WRF-Chem citations need to include those relevant to the aerosol direct and indirect effects, particularly because those processes are the focus of this paper. The standard citations for this purpose are *Chapman et al.* [2009]; *Fast et al.* [2006]; *Gustafson et al.* [2007].

p. 5, l. 12: "Resolution" needs to be changed to "grid spacing." The two are not interchangeable.

p. 5, l. 15: The authors presumably meant "grid spacing" and not "width."

p. 8, l. 4: The "p" should be subscripted.

p. 11, l. 5: I do not understand what is trying to be conveyed by "...presenting the ensemble always maximum time..." This appears to be a garbled sentence.

p. 15, l. 9: Reference to Forkel et al. (2015) is mistyped.

**References**

Chapman, E. G., W. I. Gustafson, R. C. Easter, J. C. Barnard, S. J. Ghan, M. S. Pekour, and J. D. Fast (2009), Coupling aerosol-cloud-radiative processes in the WRF-Chem model: Investigating the radiative impact of elevated point sources, *Atmos. Chem. Phys.*, *9*, 945–964, doi:10.5194/acp-9-945-2009.

Fast, J. D., W. I. Gustafson, R. C. Easter, R. A. Zaveri, J. C. Barnard, E. G. Chapman, G. A. Grell, and S. E. Peckham (2006), Evolution of ozone, particulates, and aerosol direct radiative forcing in the vicinity of Houston using a fully coupled meteorology-chemistry-aerosol model, *J. Geophys. Res.*, *111*, D21305, doi:10.1029/2005jd006721.

Gustafson, W. I., E. G. Chapman, S. J. Ghan, R. C. Easter, and J. D. Fast (2007), Impact on modeled cloud characteristics due to simplified treatment of uniform cloud condensation nuclei during NEAQS 2004, *Geophys. Res. Lett.*, *34*, L19809, doi:10.1029/2007gl0300321.

---

## Referee Comment (RC4) · Anonymous Referee #4 · 8 Feb 2017

General comments: The manuscript "Regional effects of atmospheric aerosols on temperature: an evaluation of an ensemble of on-line coupled models", focusing on the prognostic of the temperature field at 2 meters across Europe, main goal was to evaluate the performance of individual outputs from distinct simulations based on two modelling systems, WRF-CHEM and COSMO, against the performance of an ensemble based on these same individual simulations. A relevant aspect of the study is the performance assessment of both individual runs and the ensemble under different modelling context of aerosol particles effects prescription: a) neglecting any aerosol effects; b) including only aerosol-radiation interaction; c) and including aerosol-radiation plus aerosol-cloud-radiation interactions. According to the manuscript, the main conclusion

obtained from the analyses is that the inclusion of aerosol effect feedbacks did not have a significant impact on the bias observed between modelled and observed temperature. However, the spatial and temporal variability are better represented when aerosol radiative effects are included in the simulations. The subject of the manuscript is within the scope of ACP and it is a relevant scientific issue. However, there is several issues regarding the way it is addressed. Therefore, I think that the manuscript needs some work before its acceptance.

There are some points/aspects in the manuscript that I would like to comment: The manuscript has a critical issue, a fundamental element behind the scientific object discussed is the aerosol effects, in both radiation and clouds, nevertheless, the aerosol horizontal and vertical distribution loading is absolutely absent from the manuscript. To addressed the influence of these effects on the surface temperature it is crucial to have, at least, a clear notion on the aerosol horizontal loading, observed and modelled. The difference between them may be critical to understand the potential discrepancies between model and observation regarding temperature. The daily variability of the aerosol loading, at least over the regions highly affect by the fires and dust episodes would be also important to the understanding of the models performance as regard to the temporal variability of the surface temperature. That would help to better contextualize some of your conclusions.

Although the focus is on the aerosol effects feedbacks on the surface temperature, to provide a comprehensive perspective of the aerosol effect feedbacks, I would recommend the authors to describe the dominant meteorological context during the episodes, with special focus on those variables that govern the surface temperature field. Meteorological scenario (large scale patterns) is also relevant since it can enhance or mask aerosols effects feedbacks.

Though the manuscript emphasizes the ensemble performance evaluation, the modelling experiment done it is well designed and it opens many and relevant/interesting possibilities of analysis that could be explored to evaluate modelling issues and the

feedbacks induced by aerosol-radiation and aerosol-cloud-radiation prescription and, therefore, improve the manuscript discussion and results. A reason to suggest this is that, although the characterization of the uncertainty associated to the use of different modelling systems is pointed out as one of the manuscript goal, few is discussed on this matter throw out the topic of results. Being the WRF-CHEM individual models dominant and presented basically the same configuration, although from distinct institution, the ensemble results seems to resemble WFR_CHEM features, which is clear when Bias are analysed. Moreover, I wonder about the inclusion of a WRF-CHEM model version with a spatial resolution substantially higher than the others an its influence on the ensemble results. I also wonder about the effectiveness of the discussion largely based on domain-averaged values given the domain considered. As highlighted before, the experiment design provides interesting alternative to explore aerosol prescription effects if the analysis was not only focused on domain-averaged values. The simultaneous analyses of the inclusion(exclusion) of aerosol effects based on the ensemble field seems to be a challenge since the ensemble may reflect compensation between features from individual models.

Specific comments:

Page 2, Line 03: "...due to direct aerosol-radiation..." to "... due to the direct aerosol-radiation..."

Page 2, Line 04: "...from aerosol-cloud interactions..." to "... from aerosol-cloud-radiation interactions..."

Page 2, Line 09: "...and minimum temperature over Europe..." to "...and minimum temperature at 2 meters over Europe..."

Page 2, Line 10: "The evaluated model outputs originate..." to "The evaluated models outputs originate ..."

Page 2, Line 11: "The case studies cover two important..." to "The cases studies cover

two important. . ."

Page 2, Line 12-13: ". . .a heat wave and forest fires episode. . ." to ". . .a heat wave event and a forest fires episode. . ."

Page 2, Line 19: ". . .those areas closest to emissions sources. . ." to ". . .those areas closest to significant emissions sources. . ."

Page 2, Line 17: "the spatio-temporal variability and correlation coefficients are improved for the cases under study when atmospheric aerosol radiative effects are included, especially for those areas closest to emissions sources of atmospheric aerosols" How can one see that without any plot showing emissions and/or aerosol loading in the manuscript?

Page 2, Line 21 -22: "Atmospheric aerosol particles are known to have an impact on Earth's radiative budget due to their optical, microphysical and chemical properties,. . ."

to

"Atmospheric aerosol particles are known to have an impact on Earth's radiative budget due to their interaction with radiation and clouds properties, which are dependent on their optical, microphysical and chemical properties . . ."

Page 2, Line 23-26: "They influence climate by modifying both the global energy balance through absorption and scattering of radiation (direct effect), and by acting as cloud condensation nuclei, thus affecting cloud droplet size distributions and lifetime (Twomey 1977; Lohmann and Feichter, 2005; Chung, 2012) and the reflectance and persistence. . ."

to

"They influence climate by modifying the global energy balance through both absorption and scattering of radiation (direct effect) and by acting as cloud condensation nuclei, thus affecting clouds droplet size distribution, lifetime (Twomey 1977; Lohmann

and Feichter, 2005; Chung, 2012) and reflectance (indirect effects). . ."

Page 3, Line 15-17: ". . .the air quality model evaluation international initiative (AQMEII) in its phase 2 (Alapaty et al., 2012; Galmarini et al., 2015) focused on the assessment of. . ."

to

". . ., in its phase 2, the air quality model evaluation international initiative (AQMEII) (Alapaty et al., 2012; Galmarini et al., 2015) focused on the assessment of. . ."

Page 3, Line 20: ". . .aerosols, radiation, clouds, and precipitation. . ." to ". . .aerosols, radiation, clouds and precipitation. . ."

Page 3, Line 20: ". . .a coordinated exercise of Working Groups 2 and 4 of the COST Action ES1004 (EuMetChem, http://eumetchem.info) emerged, in order to take into account the radiative feedbacks, due to atmospheric aerosol effects over meteorology. . ."

to

". . ., a coordinated exercise of the working groups 2 and 4 of the COST Action ES1004 (EuMetChem, http://eumetchem.info) emerged in order to take into account the radiative feedbacks of atmospheric aerosol effects on meteorology."

Page 3, Line 26: ". . .of their strong potential of aerosol interactions. . ."

to

". . .of their strong potential for aerosol-radiation and aerosol-cloud-radiation interactions. . ."

Page 3, Line 28: ". . .onto meteorology. . ." to ". . .on meteorology. . ."

Page 4, Line 1: Specify temperature at which level (surface, 2 meters?) the paragraph is referring.

Page 4, Line 5: "Forkel et al. (2012) studied an episode in June and July. . ." specify

the nature of the episode that is discussed here.

Page 4, Line 7: ". . .this reduction was reflected in its spatial distribution of the planetary boundary layer height. . ." please, clarify.

Based on the first sentence of the last paragraph: Page 4 Line 11:"However, all these studies are based on individual model evaluations and do not take into account an ensemble of regional models, in order to build confidence on model simulations and to characterize the uncertainty associated to the use of different modelling systems"

I'm tempted to suggest another perspective on the sentence that describe the manuscript main goal (just a suggestion)

from

Page 4 Line 14:" ". . .the objective of this work is to assess whether the inclusion of aerosol radiative feedbacks during two important atmospheric aerosol episodes of the year 2010 improves the outputs of an ensemble of regional on-line coupled models for maximum, mean and minimum temperature at 2 meters over Europe."

to

". . .the objective of this work is to assess whether the outputs of an ensemble of regional on-line coupled models simulations including aerosol radiative feedbacks, during two important atmospheric aerosol episodes of the year 2010, improves the prognostic for maximum, mean and minimum temperature at 2 meters over Europe"

Page 4 Line 23: ". . .the Russian 2010 heatwave and wildfires episode in summer 2010 (25 July-15 August 2010) . . ."

to

". . .the Russian heatwave and wildfires episode in the summer of 2010 (25 July-15 August 2010). . ." Page 5 Line 1: ". . .which does not consider any feedbacks to meteorology with simulated aerosol (NRF),"

to

"...which does not consider any aerosol effects feedbacks to meteorology (NRF),"

Page 5 Line 2: "...where aerosol-cloud interactions based on simulated aerosol concentrations and direct and indirect aerosol effects are considered (ARI+ACI)." I think this sentence needs to be improved.

Page 5 Line 11-12: "...with different chemistry and physics options and performed episodes..." The last part of the sentence "and performed episodes" did not make sense to me, please, clarify.

Page 5 Line 14: "...grid with of 0,125deg /approximately 14 km) there is an additional..." to "...grid with of 0.125 deg (approximately 14 km) and there is an additional..."

Page 5 Line 21: "...uses the Model for Simulating Aerosol Interactions and Chemistry (MOSAIC)(4 bins)..." Please, provide the meaning of "(4 bins)".

Page 6 Line 3: "2.2 Emissions and boundary conditions" The topic is mentioning "boundary conditions" but it only describe emissions sources. How about dust emission, since one of the case study focus is on an event of dust transport.

Page 6 Line 9: "...volatile organic com- pounds..." to "...volatile organic compounds..."

Page 6 Line 25: "...alter significantly any of our results..." to "...alter significantly our results..."

---

## Referee Comment (RC5) · Anonymous Referee #5 · 14 Feb 2017

The manuscript analyzes the performance of on-line coupled models in forecasting air temperature near the surface over Europe including the aerosol direct and indirect effects. Output fields of mean, maximum and minimum air temperatures from individual models and in ensemble are compared to observations. Overall, the manuscript is well written and can contribute to improve weather forecast models. Before acceptance, though, some questions must be addressed, as follow:

1) Since the effect on air temperature due to aerosol is rather a feedback effect, it would be interesting to evaluate the models skills in reproducing AOD and other aerosol optical properties (particularly single scattering albedo), for the direct effect and the cloud field for the aerosol-cloud-interactions. Also, under background (low AOD) and cloudless sky conditions how the models output compare with the observations? Such analyses will help identify if other effects rather than those caused by aerosol interactions are of higher significance in affecting air temperature near the surface.

2) Even though Jiménez-Guerrero is one of the co-authors of the manuscript and reference to Jiménez-Guerrero et al. (2013) is given, at the validation methodology section (page 7, from line 7 and following), many parts of the text are identical to the reference, what must be avoided.

3) Could you please explain why the models performance can be different in reproducing mean, maximum and minimum air temperature, particularly considering the aerosol effects?

4) When including the aerosol feedbacks, was it possible to observe better model performance in regions of higher AOD?

Technical corrections: Page 5, line 14: please replace "grid with of 0,125deg" by "grid with 0.125deg" (notice the use of comma as decimal separator);

Page 12, line 20: please replace "intermediate representation the variability" by "intermediate representation of the variability";

Page 15, line 11: please correct Forkel et al. 2015.

---

## Author Comment (AC1) · 5 May 2017

Q: [R1] […] The article has a very clear objective of evaluating the simulations of coupled models including aerosol interactions with radiation and clouds with respect to surface temperatures. This is clearly of general interest since it is an impact on weather forecasting that may motivate changes in operational models around the world. The improvement of temporal variability is an important result. [R2]: […] The results points towards an improvement of spatial and temporal correlations when adding these interactions. The paper is well written and the topic is in the scope of the Journal. [R3] […] The paper basically shows that including aerosol effects generally leads to a small improvement in the simulated surface temperature. [R4]: […] According to the manuscript, the main conclusion obtained from the analyses is that the inclusion of aerosol effect feedbacks did not have a significant impact on the bias observed between modelled and observed temperature. However, the spatial and temporal variability are better represented when aerosol radiative effects are included in the simulations. The subject of the manuscript is within the scope of ACP and it is a relevant scientific issue. [R5]: […] Overall, the manuscript is well written and can contribute to improve weather forecast models.

A: First of all, we would like to thank the reviewers for their positive opinion on the paper and its importance and their very valuable comments. All the reviewers raise very interesting points, which are addressed point-by-point below. (Reviewer's comments are displayed in black, replies in blue fonts).

Q: [R1] There is a lack of discussion on how well the aerosol concentrations for the two episodes compare with actual observations […] In the cases used in this paper, how do these values compare to station data? [R2] It would be good to also provide a figure(s) showing observed AOD and clouds for the period of analysis and then how each model was able to represent them. If this is performed in other paper please include it anyways for context. [R3] […] How can one see that without any plot showing emissions and/or aerosol loading in the manuscript? [R4] The aerosol horizontal and vertical distribution loading is absolutely absent from the manuscript. To address the influence of these effects on the surface temperature it is crucial to have, at least, a clear notion on the aerosol horizontal loading, observed and modelled. [R5] It would be interesting to evaluate the models skills in reproducing AOD and other aerosol optical properties.

A: We fully agree with the reviewers' comments. In fact, AOD representation is so important that the model representation and evaluation of aerosols and AOD are presented in a full accompanying paper, which will be soon re-submitted to this same issue of ACP after considering its own reviewer's

comments (*Palacios-Peña et al., submitted. An assessment of aerosol optical properties from remote sensing observations and an ensemble of regional chemistry-climate coupled models over Europe*).

However, a short description of the model behaviour for AOD representation is included in the manuscript at the beginning of the results section.

Q: [R1] The reader is left without means to judge whether this is a convincing case or not. In the conclusions, the authors reinforce this feeling by saying that this evaluation should be performed for cases with "episodes with stronger effects on the aerosol cloud interactions" and mentioning that in one of the cases larger concentration were found over the Mediterranean Sea where the evaluation in not performed.

A: The reviewer is right. Despite the substantiated election of the cases in different parts of the bibliography (especially, those related to EuMetChem), the sentence stated by the reviewers is an unfortunate claim. Our intention was to highlight that the ARI+ACI interactions are more pronounced in this episode over ocean areas (unluckily not covered by E-OBS). So we have rephrased the sentence in the conclusions for a better clarification:

*"In order to further investigate the impact of including the aerosol interactions in online coupled models, more episodes with effects on the aerosol-radiation-cloud interactions should be considered. In this work, the fires episode represents a situation of clear skies, and therefore the ARI feedbacks are dominant. The dust episode election permits to study aerosol-cloud interaction, most of the ARI+ACI differences found in the models with respect to the base case were found over the Mediterranean sea. Since the observational data E-OBS only has values over land, the effect of ARI+ACI were not evaluation here. Unfortunately part of the interpretation of the results may be missed due to the unavailability of this database over the ocean."*

Q: [R1] Page 6, line 8 you mention annual emissions, what is actually used in a daily/hourly basis? Are these cases significant from the point of view of high emissions and concentration of aerosol over land?

A: The methodology for emissions follows that explained by Im et al. (2015). As stated here, consistent temporal profiles (diurnal, day-of-week, seasonal) and vertical distributions were also made available to AQMEII and EuMetChem participating groups for time disaggregation. The temporal profiles for the EU anthropogenic emissions were provided from Schaap et al. (2005). This information has been clarified in the revised manuscript.

Im, U., Bianconi, R., Solazzo, E., Kioutsioukis, I., Badia, A., Balzarini, A., Baró, R., Bellasio, R., Brunner, D., Chemel, C., Curci, G., Flemming, J., Forkel, R., Giordano, L., Jiménez-Guerrero, P., Hirtl, M., Hodzic, A., Honzak, L., Jorba, O., Knote, C., Kuenen, J. J., Makar, P. A., Manders-Groot, A., Neal, L., Pérez, J. L., Pirovano, G., Pouliot, G., Jose, R. S., Savage, N., Schroder, W., Sokhi, R. S., Syrakov,

D., Torian, A., Tuccella, P., Werhahn, J., Wolke, R., Yahya, K., Zabkar, R., Zhang, Y., Zhang, J., Hogrefe, C., and Galmarini, S.: Evaluation of operational on-line-coupled regional air quality models over Europe and North America in the context of AQMEII phase 2. Part I: Ozone, Atmospheric Environment, 115, 404–420, 2015.

Schaap, M., Roemer, M., Sauter, F., Boersen, G., Timmermans, R., Builtjes, P.J.H., 2005. LOTOS-EUROS: Documentation. TNO report B&O-A, 2005-297, Apeldoorn.

Q: [R1] Pages 7, Equation 2, objectively define ˆ ; what is the operation defined in eq. 2? What is Vick ? [R2] Equation 2. There is either an error on the notation or a variable is not defined (Vˆk_ic)

A: We strongly appreciate the reviewer's suggestions. There was a typo in the definitions of the equations of variability that has been corrected in the revised version of the manuscript.

Q: [R1] Page 9, lines 3-5, for the whole period, 60 days in one case and 30 days in the other case? Only in figure 10 the exact period is mentioned. Please state that in the text.

A: The exact period is specified in pag 4, lines 23 to 25

Q: [R1] The numbers that summarize results should be organized in tables so they can be easily compared. [R3] The figures with maps are presented in such a way that one cannot gain a clear quantitative understanding of how the models differ beyond very large differences. […] In most cases, the maps look identical without extremely close examination. A better way needs to be found to present this information. [R1] Page 9, lines 6-28 the average numbers of the bias for each case and run should be summarized in a Table.

A: Following the reviewer's advice, a Table summarizing all the results has been including in the revised version of the manuscript (Tables 2 and 3).

Q: [R1] […] What is needed is a way to represent the effect on temperature in places where there is a high concentration of aerosol. You could choose a given simulation time with very high aerosol concentrations and show model performance for temperatures. [R2] […] I suggest the authors to restrict their analysis only to regions where large aerosol impact is expected. For instance, bias could be computed only for regions where AOD is over a fixed threshold, or you could weigh the bias by AOD. [R4] I also wonder about the effectiveness of the discussion largely based on domain-averaged values given the domain considered.

A: The domains presented here are sub-domains of the ensemble of EuMetChem simulations, which covers a European-wide domain. The elections of these sub-domains was based precisely on those European sub-area where the aerosol could affect most the meteorological variables. It is for this reason that these sub-domains for the wildfires episode and the dust episode were selected. Therefore, in our opinion, results regarding bias or

correlation would not change importantly if different domains were selected.

Q: [R1]: [...] There is very little discussion on why given models may perform better than others.

A: We fully agree with the reviewer's comments. However, the point raised by the reviewers was already addressed in Brunner et al. (2015), where the authors present an operational analysis of model performance with respect to key meteorological variables relevant for atmospheric chemistry processes and air quality. So the reader is referred to that work for clarification.

Q: [R3] Gridded temperature data sets typically show a range of uncertainties due to different methodologies used to spatially distribute and average point observations. [...] Most likely, the differences identified between running the models with and without aerosol feedbacks is smaller than the differences between observation data sets.

A: The reviewer raises a very interesting point. One of the works coauthored the corresponding author of this manuscript (Gómez-Navarro et al., 2012) examines to what extent the evaluation and ranking of an ensemble of regional climate models, according to their ability to reproduce the observed climatologies, is sensitive to the choice of the reference observational data set. The authors found that for maximum and minimum temperatures, it turns out that uncertainties among observations are at least as relevant as uncertainties among the models within an ensemble.

However, the main objective of this work is not to rank the ensemble of simulations included in EuMetChem, but to provide a comprehensive comparison between simulations and to assess the differences when including the aerosol feedbacks. In this sense, the E-OBS dataset was selected because of its wide use in scientific literature when evaluating regional climate models.

Gómez-Navarro, J. J., J. P. Montávez, S. Jerez, P. Jiménez-Guerrero, and E. Zorita (2012), What is the role of the observational dataset in the evaluation and scoring of climate models?, Geophys. Res. Lett., 39, L24701, doi:10.1029/2012GL054206.

Q [R3]: The "third" configuration needs to be defined in relation to the handling of aerosol and/or cloud droplet assumptions when aerosol-cloud interactions are disabled. [...] However, one cannot run a model without aerosol and still form clouds, since the aerosols are required for forming cloud droplets in almost all physically relevant conditions. So, even "without" aerosol-cloud interactions there are still significant assumptions built into the models to account for the ACI processes.

A: The reviewer is right. This point has been clarified in the manuscript (Second paragraph, Section 2) as follows: *"Although NRF case does not*

*consider the aerosol effects and feedbacks, there is a standard aerosol assumption of some continental aerosol (250 cm$^{-3}$ used by WRF-Chem in the absence of ACI for estimating cloud droplet number). On the other hand, ARI uses this constant value for accounting the interaction between aerosols and clouds, but allows the modification of the radiation budget by using the on-line estimated aerosols. Last, the ARI+ACI cases are based on simulated aerosol concentrations, which interact both with radiation and aerosols. The common setup for the participating models and a unified output strategy allow analyzing the model output with respect to similarities and differences in the model response to the aerosol direct effect and aerosol-cloud interactions."*

Q: [R4] I would recommend the authors to describe the dominant meteorological context during the episodes, with special focus on those variables that govern the surface temperature field.

A: The weather conditions during the Russian forest fires were mainly dry and particularly hot, with light winds. During this situation, the sea-level pressure (SLP) showed a high-pressure system over the northeast part of the Russian area, finding a strong positive SLP anomaly for this period. This resulted in a strong positive surface temperature anomaly accompanied by weak winds from the southeast. On the other hand, a very deep trough characterizes the dust period situation with a vortex reaching 20 degrees of north latitude. This situation is maintained for several days, causing a continuous transport in middle levels. It is also worth mentioning the blocking situation over all central Europe. The dust event was dominated by strong south-easterly wind. This may explain windblown dust emissions increasing with wind speed and being transported to some parts of the European area.

This description has been included in the revised version of the manuscript, Section 2.

Q: [R4] Being the WRF-CHEM individual models dominant and presented basically the same configuration, although from distinct institution, the ensemble results seems to resemble WFR_CHEM features, which is clear when Bias are analysed. Moreover, I wonder about the inclusion of a WRF-CHEM model version with a spatial resolution substantially higher than the others and its influence on the ensemble results. […] The simultaneous analyses of the inclusion(exclusion) of aerosol effects based on the ensemble field seems to be a challenge since the ensemble may reflect compensation between features from individual models.

A: The meteorological variables simulated by regional models suffer from uncertainties arising from a variety of sources such as internal variability, different model formulations, etc. Results found in the literature indicate that the ensemble mean is usually less biased than the individual members (Fernández et al. 2009, Knutti et al. 2010; Kjellström et al., 2011).

As stated by Annan and Hargreaves (2011), one hypothesis for the improvement of the ensemble mean when compared to the performance of the individual models is the paradigm of models being considered as independent samples from some distribution that is centered on the truth, as in this case the ensemble mean could be expected to converge to the truth as more models are added to the ensemble.

With respect to WRF-CHEM individual models being dominant, Jerez et al. (2013) indicate that the uncertainties associated to the physics of the driving meteorological model are of the same order of magnitude as the uncertainties associated with a multi-model ensemble. Therefore, even though WRF-CHEM models are dominant in the ensemble, the diversity of the parameterizations elected make the election feasible for an ensemble analysis.

Annan JD, Hargreaves JC (2010) Reliability of the CMIP3 ensemble. Geophys Res Lett 37:L02703, doi:10.1029/2009 GL041994

Fernández J, Primo C, Cofiño AS, Gutiérrez JM, Rodríguez MA (2009) MVL spatiotemporal analysis for model inter- comparison in EPS: application to the DEMETER multi- model ensemble. Clim Dyn 33:233–243

Knutti R, Furrer R, Tebaldi C, Cermak J, Meehl GA (2010) Challenges in combining projections from multiple cli- mate models. J Clim 23:2739–2758

Kjellström E, Nikulim F, Hanson U, Strandberg G, Ullerstig A (2011) 21st century changes in the European climate: uncertainties derived from an ensemble of regional cli- mate model simulations. Tellus 63A:24–40

Q: [R5] Even though Jiménez-Guerrero is one of the co-authors of the manuscript and reference to Jiménez-Guerrero et al. (2013) is given, at the validation methodology section (page 7, from line 7 and following), many parts of the text are identical to the reference, what must be avoided.

A: An effort has been made to rewrite this part of the manuscript, despite keeping the same nomenclature for the statistical figures.

**MINOR COMMENTS:**

Q: [R1] Page 4, lines 20-23 – please correct sentence structure

A: These lines have been corrected in the revised version

Q: [R1] Page 35 – Table 1 – define CS1, CS2, DE3, ES1, ES3, in the text you use things like, C11, C12...etc, this should be defined in the text.

A: This information has been clarified in the revised manuscript.

Q: [R1] Page 14, lines 23-26 – for Tmin the case ENS-C13 is perhaps an exception?

A: The reviewer is right. This comment has been introduced in the revised version of the manuscript.

Q: [R2] Page 2, line 10. Somewhere in the text define "EuMetChem COST Action ES1004"

A: EuMetChem stands for "European framework for online integrated air quality and meteorology modelling". This definition has been introduced in the second paragraph of the Introduction.

Q: [R2] Page 2, Line 19. I think that the statement "especially for those areas closest to emissions sources of atmospheric aerosols" is not explained or mentioned in the text.

A: This sentence has been removed from the revised version of the abstract.

Q: [R2] Section 3.2. What strikes me the most from these results are the differences between the smoke and the dust case. For the smoke case, most of the changes come from the radiation effects, while for the dust effects most changes come from aerosol-cloud effects. Can you explain why this happens? Does the dust case has more clouds over the domain? Is it related to better cirrus representation?

A: During the fire episode, a predominantly clear-sky situation was found; therefore, the aerosol effects governing the changes in temperature are related to the aerosol-radiation effect. Conversely, during the dust episode, formation of clouds is enhanced because of the meteorological situation explained above.

Q: [R2] Page 4, line 22. Erase "are run"

A: Line has been corrected

Q: [R2] Page 15, line 11. Fix the forkeletal2015 reference. [R3] p. 15, l. 9: Reference to Forkel et al. (2015) is mistyped. [R5] Page 15, line 11: please correct Forkel et al. 2015

A: Reference has been corrected in the revised version of the manuscript.

Q: [R3] p. 4, l. 27: "Three different cases" is better phrased as "Three different configurations." "Cases" implies different dates and "configurations" is more specific to what is being described.

A: This comment has been introduced in the revised version of the manuscript as suggested.

Q: [R3] p. 5, l. 10: The WRF-Chem citations need to include those relevant to the aerosol direct and indirect effects, particularly because those processes are the focus of this paper. The standard citations for this purpose are Chapman et al. [2009]; Fast et al. [2006]; Gustafson et al. [2007].

A: These references have been added to the WRF-Chem citation.

Chapman, E. G., W. I. Gustafson, R. C. Easter, J. C. Barnard, S. J. Ghan, M. S. Pekour, and J. D. Fast (2009), Coupling aerosol-cloud-radiative processes in the WRF-Chem model: Investigating the radiative impact of elevated point sources, Atmos. Chem. Phys., 9, 945–964, doi:10.5194/acp-9-945-2009.

Fast, J. D., W. I. Gustafson, R. C. Easter, R. A. Zaveri, J. C. Barnard, E. G. Chapman, G. A. Grell, and S. E. Peckham (2006), Evolution of ozone, particulates, and aerosol direct radiative forcing in the vicinity of Houston using a fully coupled meteorology-chemistry-aerosol model, J. Geophys. Res., 111, D21305, doi:10.1029/2005jd006721.

Gustafson, W. I., E. G. Chapman, S. J. Ghan, R. C. Easter, and J. D. Fast (2007), Impact on modeled cloud characteristics due to simplified treatment of uniform cloud condensation nuclei during NEAQS 2004, Geophys. Res. Lett., 34, L19809, doi:10.1029/2007gl0300321

Q: [R3] p. 5, l. 12: "Resolution" needs to be changed to "grid spacing." The two are not interchangeable.

A: It has been changed as suggested.

Q: [R3] p. 5, l. 15: The authors presumably meant "grid spacing" and not "width."

A: It has been changed as suggested.

Q: [R3] p. 8, l. 4: The "p" should be subscripted.

A: The reviewer is right. Letter "p" has been subscripted in the revised version of the manuscript.

Q: [R3] p. 11, l. 5: I do not understand what is trying to be conveyed by "...presenting the ensemble always maximum time..." This appears to be a garbled sentence.

A: The sentence has been corrected as "presenting the ensemble always maximum values for…"

Q: [R4] Page 2, Line 03: "...due to direct aerosol-radiation..." to "...due to the direct aerosol-radiation..."

A: This sentence has been changed in the revised version of the manuscript.

Q: [R4] Page 2, Line 04: "...from aerosol-cloud interactions..." to "...from aerosol-cloud-radiation interactions..."

A: This sentence has been changed in the revised version of the manuscript.

Q: [R4] Page 2, Line 09: "...and minimum temperature over Europe..." to "...and minimum temperature at 2 meters over Europe..."

A: This comment has been changed in the revised version of the manuscript.

Q: [R4] Page 2, Line 10: "The evaluated model outputs originate..." to "The evaluated models outputs originate..."

A: Changed as suggested.

Q: [R4]: Page 2, Line 11: "The case studies cover two important..." to "The cases studies cover two important..."

A: Changed as suggested.

Q: [R4] Page 2, Line 12-13: "...a heat wave and forest fires episode..." to "...a heat wave event and a forest fires episode..."

A: Changed as suggested.

Q: [R4] Page 2, Line 19: "...those areas closest to emissions sources..." to "...those areas closest to significant emissions sources..."

A: This sentence has been changed in the revised version of the manuscript.

Q: [R4] Page 2, Line 21 -22: "Atmospheric aerosol particles are known to have an impact on Earth's radiative budget due to their optical, microphysical and chemical properties,..." to "Atmospheric aerosol particles are known to have an impact on Earth's radiative Budget due to their interaction with radiation and clouds properties, which are dependent on their optical, microphysical and chemical properties..."

A: This sentence has been changed as suggested in the revised version of the manuscript.

Q: [R4] Page 2, Line 23-26: "They influence climate by modifying both the global energy balance through absorption and scattering of radiation (direct effect), and by acting as cloud condensation nuclei, thus affecting cloud droplet size distributions and lifetime (Twomey 1977; Lohmann and Feichter, 2005; Chung, 2012) and the reflectance and persistence..." to "They influence climate by modifying the global energy balance through both absorption and scattering of radiation (direct effect) and by acting as

cloud condensation nuclei, thus affecting clouds droplet size distribution, lifetime (Twomey 1977; Lohmann and Feichter, 2005; Chung, 2012) and reflectance (indirect effects)..."

A: This sentence has been changed as suggested in the revised version of the manuscript.

Q: [R4] Page 3, Line 15-17: " ...the air quality model evaluation international initiative (AQMEII) in its phase 2 (Alapaty et al., 2012; Galmarini et al., 2015) focused on the assessment of ..." to "..., in its phase 2, the air quality model evaluation international initiative (AQMEII) (Alapaty et al., 2012; Galmarini et al., 2015) focused on the assessment of ..."

A: This sentence has been changed as suggested in the revised version of the manuscript.

Q: [R4] Page 3, Line 20: "...aerosols, radiation, clouds, and precipitation..." to " ...aerosols, radiation, clouds and precipitation..."

A: Changed as suggested.

Q: [R4] Page 3, Line 20: "...a coordinated exercise of Working Groups 2 and 4 of the COST Action ES1004 (EuMetChem, http://eumetchem.info) emerged, in order to take into account the radiative feedbacks, due to atmospheric aerosol effects over meteorology..." to "..., a coordinated exercise of the working groups 2 and 4 of the COST Action ES1004 (EuMetChem, http://eumetchem.info) emerged in order to take into account the radiative feedbacks of atmospheric aerosol effects on meteorology."

A: Changed as suggested.

Q: [R4] Page 3, Line 26: "...of their strong potential of aerosol interactions..." to" ... of their strong potential for aerosol-radiation and aerosol-cloud-radiation interactions ..."

A: This comment has been changed in the revised version of the manuscript.

Q: [R4] Page 3, Line 28: "...onto meteorology..." to "...on meteorology..."

A: Changed as suggested.

Q: [R4] Page 4, Line 1: Specify temperature at which level (surface, 2 meters?) the paragraph is referring.

A: Forkel et al. (2015) refer to 2-m temperature. This has been clarified in the revised version of the manuscript.

Q: [R4] Page 4, Line 5: "Forkel et al. (2012) studied an episode in June and July..." specify the nature of the episode that is discussed here.

A: The reason for selecting this two-month episode was clarified in the manuscript.

Q: [R4] Page 4, Line 7: "...this reduction was reflected in its spatial distribution of the planetary boundary layer height..." please, clarify.

A: This sentence has been rewritten in the revised version of the manuscript.

Q: [R4] Based on the first sentence of the last paragraph: Page 4 Line 11:"However, all these studies are based on individual model evaluations and do not take into account an ensemble of regional models, in order to build confidence on model simulations and to characterize the uncertainty associated to the use of different modelling systems". I'm tempted to suggest another perspective on the sentence that describe the manuscript main goal (just a suggestion) from:

Page 4 Line 14:" "...the objective of this work is to assess whether the inclusion of aerosol radiative feedbacks during two important atmospheric aerosol episodes of the year 2010 improves the outputs of an ensemble of regional on-line coupled models for maximum, mean and minimum temperature at 2 meters over Europe."

To " ...the objective of this work is to assess whether the outputs of an ensemble of regional on-line coupled models simulations including aerosol radiative feedbacks, during two important atmospheric aerosol episodes of the year 2010, improves the prognostic for maximum, mean and minimum temperature at 2 meters over Europe"

A: We strongly appreciate the reviewer's suggestion, which clarifies the objective of this contribution. The reviewer's comment has been introduced in the revised version of the manuscript.

Q: [R4] Page 4 Line 23: "...the Russian 2010 heatwave and wildfires episode in summer 2010 (25 July-15 August 2010)..." to "...the Russian heatwave and wildfires episode in the summer of 2010 (25 July-15 August 2010)..."

A: Changed as suggested.

Q: [R4] Page 5 Line 1: "...which does not consider any feedbacks to meteorology with simulated aerosol (NRF)," to "...which does not consider any aerosol effects feedbacks to meteorology (NRF),"

A: Changed as suggested.

Q: [R4] Page 5 Line 2: "...where aerosol-cloud interactions based on simulated aerosol concentrations and direct and indirect aerosol effects are considered (ARI+ACI)." I think this sentence needs to be improved.

A: This sentence has been rewritten for the sake of clarity in the revised version of the manuscript.

Q: [R4] Page 5 Line 11-12: "...with different chemistry and physics options and performed episodes..." The last part of the sentence "and performed episodes" did not make sense to me, please, clarify.

A: This sentence has been rewritten for the sake of clarity in the revised version of the manuscript.

Q: [R4] Page 5 Line 14: "...grid with of 0,125deg /approximately 14 km) there is an additional..." to "...grid with of 0.125 deg (approximately 14 km) and there is an additional..." [R5] Page 5, line 14: please replace "grid with of 0,125deg" by "grid with 0.125deg" (notice the use of comma as decimal separator).

A: Changed as suggested.

Q: [R4] Page 5 Line 21: "...uses the Model for Simulating Aerosol Interactions and Chemistry (MOSAIC)(4 bins) ..." Please, provide the meaning of "(4 bins)".

A: In order to clarify it, "4 bin" has been replaced by "4 aerosol size bins" since the term bins refers to the number of aerosol size bins considered for representing aerosol distribution with respect to their diameter.

Q: [R4] Page 6 Line 3: "2.2 Emissions and boundary conditions" The topic is mentioning "boundary conditions" but it only describe emissions sources. How about dust emission, since one of the case study focus is on an event of dust transport.

A: We appreciate the reviewer's suggestion. The description of boundary conditions has been added to the revised version of the manuscript.

Q: [R4] Page 6 Line 9: "...volatile organic compounds..." to "...volatile organic compounds..."

A: Changed as suggested.

Q: [R4] Page 6 Line 25: "...alter significantly any of our results..." to "... alter significantly our results..."

A: Changed as suggested.

Q: [R5] Page 12, line 20: please replace "intermediate representation the variability" by "intermediate representation of the variability";

A: Changed as suggested.

---

## Referee Report (RR1)

**Regional effects of atmospheric aerosols on temperature: an evaluation of an ensemble of on-line coupled models**

**for Atmospheric Chemistry and Dynamics, May 2017**

The authors have generally addressed some concerns raised during the first review and have conveniently neglected to mention other concerns. The response to the first review is organized by blending comments from each reviewer by theme, since many of the comments were common between reviewers. However, in doing this, it makes it hard for the editor to see what has and has not been addressed. The authors may choose to not revise the manuscript if they disagree with reviewer suggestions, but they should at least say why.

**Unaddressed Major Comments from this Reviewer from First Review**

The paper could be greatly strengthened by looking at why the model results improve, or at least by providing additional information to help readers gain context. This could be done by looking at the energy budgets. The benefit of having the range of models in the intercomparison is that one can examine if the aerosol-related improvements occur for the same reason in each model, or if there are compensating effects that lead to interesting nonlinearities.

Another way the paper could be improved is by looking beyond daily values. Only looking at daily values hides a lot of model deficiencies. Comparing the models against hourly temperature data, as well as moisture and PM2.5 amounts, would provide much more detail for understanding why the models change when including aerosol feedbacks. This would also bring the plume behavior of local aerosol sources more into play.

**Further Comments from Addressed Suggestions/Critiques**

The concern was raised that the authors compare the models to one gridded temperature dataset, and differences between datasets could be bigger than the differences shown due to the aerosol impacts within the models. The author confirms this problem, and even goes so far as noting that he wrote a paper showing just this fact. No modifications were made in the manuscript and the response goes on to say that "the main objective of this work is not to rank the ensemble of simulations included in EuMetChem, but to provide a comprehensive comparison between simulations…" This is an OK objective. Unfortunately, the text in the manuscript, at the end of the introduction at lines 18ff, claims an objective that makes the accuracy of the observations paramount: "the objective of this work is to assess whether the outputs of an ensemble of regional on-line coupled models simulations including aerosol radiative feedbacks… improves the prognostic for maximum, mean and

minimum temperature at 2 meters over Europe." If the authors do not wish to add a comparison with a second dataset to take observation uncertainty into account, they should at least add text to the manuscript that puts the observations into context and note the limitation of the current study due to the single dataset.

The authors attempted to clarify the issue of what it means to turn aerosol-cloud interactions on and off within a model. Text has been added on p. 5 stating "Although NRF case does not consider the aerosol effects and feedbacks, there is a standard aerosol assumption of some continental aerosol (250 cm$^{-3}$ used by WRF-Chem in the absence of ACI for estimating cloud droplet number)." However, as phrased this is a bit confusing. Please clarify. The sentence talks about aerosol assumptions and then provides a cloud droplet number concentration. The main issue is that the assumptions regarding physical mechanisms and sources of variability change between the two model configurations. The authors should double check with each modeling group using WRF to identify how they chose to not have aerosol-cloud interactions. This can either be done by not compiling in "chemistry mode" and then one gets the 250 cm$^{-3}$ droplet number concentration for Morrison microphysics. Or, one can compile with the chemistry mode turned on but not use an aerosol module. The latter sets a constant aerosol number concentration (*naer* in the namelist) instead of a cloud droplet concentration. This is important because the physical processes related to activation and cloud formation change depending on the mode used.

p. 13, l. 16–19: The authors claim the following sentence has been corrected, but it still does not make sense: "In general, coefficients of determination are highest for mean temperature (0.60 to 0.78) and lowest for minimum temperature (0.50 to 0.56), presenting the ensemble always maximum values for $\rho^2$ (0.75, 0.79 and 0.61, respectively for maximum, mean and minimum temperature)." It is unclear what is meant by "presenting the ensemble always maximum values."

Figure 1 is blurred and unreadable.

**Minor Comments**

p. 4, l. 19: on-line coupled model simulations

p. 5, l. 20: Although the NRF case

p. 10, l. 6: have a notion of the aerosol loading (it would be better to reword to not use the colloquial phrase "have a notion" and replace it with "have an understanding of")

p. 10, l 8ff: The sentence starting with "Despite the work of Palocios-Pena et al." is phrased poorly. It would be better to refer to the other work for full details and to say the current article provide brief details for context.

---

## Author Response (AR2)

First of all, we would like to thank again the reviewers for their time spent on reviewing our manuscript and their comments helping us improving it. Please see below our point-by-point replies (reviewer's comments are displayed in black, our replies in blue font).

**#Reviewer 1**

The authors have generally addressed some concerns raised during the first review and have conveniently neglected to mention other concerns […] The authors may choose to not revise the manuscript if they disagree with reviewer suggestions, but they should at least say why.

The paper could be greatly strengthened by looking at why the model results improve, or at least by providing additional information to help readers gain context. This could be done by looking at the energy budgets.

A: We fully agree with the reviewer. This work relies on a collaborative group of simulations. For taking a look at the energy balance, we would need a number of variables that were not (unfortunately) stored by the contributing groups to this simulations.

Another way the paper could be improved is by looking beyond daily values. Only looking at daily values hides a lot of model deficiencies. Comparing the models against hourly temperature data, as well as moisture and PM2.5 amounts, would provide much more detail for understanding why the models change when including aerosol feedbacks. This would also bring the plume behavior of local aerosol sources more into play.

A: The reviewer is right. E-OBS database included only daily data. Further discussion on the observational database is presented below. As we really think the reviewer comment is very appropriate, we have included hourly evaluation in a manuscript that was about to be submitted to ACP (discussion of AOD, PM2.5 evaluation, by using hourly databases), whose submission will wait for the inclusion of hourly data.

The concern was raised that the authors compare the models to one gridded temperature dataset, and differences between datasets could be bigger than the differences shown due to the aerosol impacts within the models […] If the authors do not wish to add a comparison with a second dataset to take observation uncertainty into account, they should at least add text to the manuscript that puts the observations into context and note the limitation of the current study due to the single dataset.

A: The reviewer is right. We have clarified that in the manuscript. An effort has also been made to change the text just to highlight the fact that we have used only one dataset and its limitations. Section "Observational database" has been changed as follows:

*The comparison of regional models with gridded datasets has to be carefully taken into account given the differences between available databases. For instance, Gómez-Navarro et al. (2012) showed that even in areas covered by dense monitoring networks, uncertainties in the observations are comparable to the uncertainties within state-of-the-art regional climate models, at least when they are driven by nominally perfect boundary conditions like reanalysis.*

*This work uses the E-OBS (Haylock et al., 2012) version 11.0 gridded observational database for maximum, mean and minimum temperature. E-OBS is a high-resolution European land-only daily gridded data set covering the period 1950-2014. The E-OBS 0.25 degrees regular latitude-longitude grid has been used as the reference for validation. Thus, data from all model runs have been bilinearly interpolated onto the E-OBS grid. Since the resolution of the models is similar to that of E-OBS, the interpolation procedure is not expected to alter significantly our results.*

*The election of this gridded dataset is based on the abundant scientific literature using E-OBS for the evaluation of regional climate models (e.g. Costa et al., 2012, Jiménez-Guerrero et al., 2013; Turco et al ., 2013; Ceglar et al., 2014), among many others). However, several authors highlight the E-OBS limitations. In this sense, Kysely and Plavcova (2010} compare E-OBS and a data set gridded onto the same grid from a high-density network of stations in the Czech Republic (GriSt), finding that large differences existed between the two gridded data sets, particularly for minimum temperatures and diurnal temperature range. The errors tended to be larger in tails of the distributions. Therefore, when evaluating regional models against one gridded dataset, results have been to be carefully taken into account.*

The authors attempted to clarify the issue of what it means to turn aerosol-cloud interactions on and off within a model. Text has been added on p. 5 [..] However, as phrased this is a bit confusing. Please clarify. The sentence talks about aerosol assumptions and then provides a cloud droplet number concentration. [..] The authors should double check with each modeling group using WRF to identify how they chose to not have aerosol-cloud interactions. This can either be done by not compiling in "chemistry mode" and then one gets the 250 cm-3 droplet number concentration for Morrison microphysics. Or, one can compile with the chemistry mode turned on but not use an aerosol module. The latter sets a constant aerosol

number concentration (naer in the namelist) instead of a cloud droplet concentration. This is important because the physical processes related to activation and cloud formation change depending on the mode used.

A: The reviewer is right. We have checked with all WRF-Chem groups that the number we have given is done by not compiling the chemistry mode and getting 250 cm$^{-3}$ droplet number concentration for microphysics in C1X and C2X. We have rephrased our sentence to clarify that in the manuscript. So the sentence remains as:

*Although the NRF case does not consider the aerosol effects and feedbacks, this configuration considers an assumption of 250 cm$^{-3}$ used by WRF-Chem in the absence of ACI for estimating cloud droplet number. This number is used in the corresponding microphysics parameterization (Morrison or Lin).*

p. 13, l. 16–19: The authors claim the following sentence has been corrected, but it still does not make sense: "In general, coefficients of determination are highest for mean temperature (0.60 to 0.78) and lowest for minimum temperature (0.50 to 0.56), presenting the ensemble always maximum values for ρ2 (0.75, 0.79 and 0.61, respectively for maximum, mean and minimum temperature)." It is unclear what is meant by "presenting the ensemble always maximum values."

A: Here we meant that the coefficients of determination for TEMP are higher than those found for TMAX or TMIN. The lowest coefficients are estimated for TMIN when compared to the other variables. Moreover, the coefficients of determination for the ensemble are always higher when compared to the ρ$^2$ of the individual models. This is found for the 3 studied variables. We have rewritten the sentence in this sense:

*In general, coefficients of determination are highest for mean temperature (ranging from 0.60 to 0.78 depending on the individual model) with respect to minimum and maximum temperature. The variable with the lowest ρ$^2$ is minimum temperature (varying from 0.50 to 0.56 depending on the model). Moreover, the coefficient of determination for the ensemble is always higher than that of each individual models for the three studied variables (0.75, 0.79 and 0.61, respectively for maximum, mean and minimum temperature).*

Figure 1 is blurred and unreadable.

A: The reviewer is absolutely right, but this is related to the quality of the images has been reduced because of the size of the original pdf file containing highquality figures (over 200 MB). When final submission is done, the original pdf file for Figure 1 will be uploaded.

p. 4, l. 19: on-line coupled models simulations

A: Changed as suggested.

p. 5, l. 20: Although the NRF case

A: Changed as suggested.

p. 10, l. 6: have a notion of the aerosol loading (it would be better to reword to not use the colloquial phrase "have a notion" and replace it with "have an understanding of")

A: Changed as suggested.

p. 10, l 8ff: The sentence starting with "Despite the work of Palacios-Pena et al." is phrased poorly. It would be better to refer to the other work for full details and to say the current article provide brief details for context.

A: Changed as suggested.

**#Reviewer 2**

Figure 1 shows that the dust case has low values of AOD. Why is this a good case for the objective of the paper which is to show the impact on surface temperature?

A: The reviewer is right. The dust episode has not as high AOD levels at 550 nm as the Russian Forest Fires, but the aerosol loads are comparable in magnitude. Dust extinction is more noticeable at higher longwaves because of the size distribution of the dust particles. However, it does not mean that the aerosol load is low. We would have the same issue whatever dust episode we would select. The dust case was selected since it was also a humid episode with some rain over the Mediterranean and the dust plume effects extended over Europe.

The spatial panels in Figures 1 - 10 are quite difficult to read, are they really necessary, all of them?

A: The reviewer is right, in this sense we have added information with additional tables (Table 2, 3, 4 and 5) as well as information on the text. We considered that it was also interesting to see the Base case, E-OBS levels as well as differences between models.

The authors have chosen not to show specific areas where the clouds and/or plume impact surface temperatures. This would enhance the value of the article and make the point more clearly.

A: In order to follow the reviewers's suggestion, two new tables (Table 3 and Table 5) have been added to the final version of the manuscript. These tables summarize the results of those specific areas where the plume would have higher impacts on surface temperatures. For that, we have calculated the bias and the coefficient of determination by masking those timesteps and areas when 1-hr AOD550 > 1.0 for the fires episodes and AOD550 > 0.5 for the dust case. The results obtained are very similar and do not modify the previous discussion presented in the manuscript. However, because of its interest, the information has been introduced in the revised version of the manuscript.

**#Reviewer 3**

Page 2, Line 22: "…which are dependent…" to "… which is dependent…"

A: Changed as suggested.

Page 10, Line 09: "…fully compiled the AOD evaluation against diverse satellite observations of the ensemble considered in this work…" to "…fully compiled the evaluation of AOD of the ensemble considered in this work against diverse satellite observations …"

A: This sentence here has been changed following the Reviewer#1 suggestion.

Need to include a reference to MODIS AOD product (e.g. Levy et al., 2013)

A: The following reference has been added.

Levy, R., Mattoo, S., Munchak, L., Remer, L., Sayer, A., Patadia, F., and Hsu, N.: The Collection 6 MODIS aerosol products over land and ocean, Atmospheric Measurement Techniques, 6, 2989-3034, 2013.

Page 10, Line 18: "a low overestimation …" to "a lower overestimation …"

A: Changed as suggested